# Photooxidation of pinonaldehyde at ambient conditions investigated in the atmospheric simulation chamber SAPHIR

Michael Rolletter[1], Marion Blocquet[1,a], Martin Kaminski[1,b], Birger Bohn[1], Hans-Peter Dorn[1], Andreas Hofzumahaus[1], Frank Holland[1], Xin Li[1,c], Franz Rohrer[1], Ralf Tillmann[1], Robert Wegener[1], Astrid Kiendler-Scharr[1], Andreas Wahner[1], and Hendrik Fuchs[1]

[1]Institute of Energy and Climate Research, IEK-8: Troposphere, Forschungszentrum Jülich GmbH, Jülich, Germany
[a]now at: Ministère de l'Education Nationale et de la Jeunesse, 110 rue de Grenelle, 75357 Paris SP 07
[b]now at: Federal Office of Consumer Protection and Food Safety, Department 5: Method Standardisation, Reference Laboratories, Resistance to Antibiotics, Berlin, Germany
[c]now at: College of Environmental Sciences and Engineering, Peking University, Beijing, China

*Correspondence to:* Hendrik Fuchs (h.fuchs@fz-juelich.de)

**Abstract.** The photooxidation of pinonaldehyde, one product of the $\alpha$-pinene degradation, was investigated in the atmospheric simulation chamber SAPHIR under natural sunlight at low NO concentrations ($< 0.2$ ppbv) with and without an added hydroxyl radical (OH) scavenger. With scavenger, pinonaldehyde was exclusively removed by photolysis, whereas without scavenger, the degradation was dominated by reaction with OH. In both cases, the observed rate of pinonaldehyde consumption was faster than predicted by an explicit chemical model, the Master Chemical Mechanism (MCM, version 3.3.1). In the case with OH scavenger, the observed photolytic decay can be reproduced by the model, if an experimentally determined photolysis frequency is used instead of the parameterization in the MCM. A good fit is obtained, when the photolysis frequency is calculated from the measured solar actinic flux spectrum, absorption cross-sections published by Hallquist et al. (1997), and an effective quantum yield of 0.9. The resulting photolysis frequency is 3.5 times faster than the parameterization in the MCM. When pinonaldehyde is mainly removed by reaction with OH, the observed OH and hydroperoxy radical ($HO_2$) concentrations are underestimated in the model by a factor of 2. Using measured $HO_2$ as a model constraint brings modeled and measured OH concentrations into agreement. This suggests that the chemical mechanism includes all relevant OH producing reactions, but is missing a source for $HO_2$. The missing $HO_2$ source strength of $(0.8 \text{ to } 1.5)\,\mathrm{ppbv\,h^{-1}}$ is similar to the rate of the pinonaldehyde consumption of up to $2.5\,\mathrm{ppbv\,h^{-1}}$. When the model is constrained by $HO_2$ concentrations and the experimentally derived photolysis frequency, the pinonaldehyde decay is well represented. The photolysis of pinonaldehyde yields $0.18 \pm 0.20$ formaldehyde molecules at NO concentrations of less than $200\,\mathrm{pptv}$, but no significant acetone formation is observed. When pinonaldehyde is also oxidized by OH under low NO conditions (maximum $80\,\mathrm{pptv}$), yields of acetone and formaldehyde increase over the course of the experiment from 0.2 to 0.3 and from 0.15 to 0.45, respectively. Fantechi et al. (2002) proposed a degradation mechanism based on quantum-chemical calculations, which is considerably more complex than the MCM scheme and contains additional reaction pathways and products. Implementing these modifications results in a closure of the model-measurement discrepancy for the products acetone and formaldehyde, when pinonaldehyde is degraded only by photolysis. In contrast, the underprediction of formed acetone and formaldehyde is worsened compared to model results by the MCM,

when pinonaldehyde is mainly degraded in the reaction with OH. This shows that the current mechanisms lack acetone and formaldehyde sources for low NO conditions like in these experiments. Implementing the modifications suggested by Fantechi et al. (2002) does not improve the model-measurement agreement of OH and $HO_2$.

# 1 Introduction

Globally, emissions of biogenic non-methane volatile organic compounds (NMVOCs) in the atmosphere are ten times higher than emissions of anthropogenic NMVOCs (Guenther et al., 2012). Of these emissions, monoterpenes (C10-compounds) represent approximately 15 % (Guenther et al., 2012) of the total emissions. Monoterpenes are mainly oxidized in the atmosphere by ozonolysis or their reaction with the hydroxyl radical (OH) during daytime. However, also oxidation by the nitrate radical ($NO_3$) during nighttime can be of importance enhanced by nocturnal monoterpene emissions (Calogirou et al., 1999; Atkinson and Arey, 2003). Oxidation products significantly contribute to the global production of, for example, acetone (Jacob et al., 2002). In addition, low-volatile organic oxidation products play an important role for the formation of secondary organic aerosol (SOA) (Kanakidou et al., 2000). The oxidation of monoterpenes and their oxidation products is also of importance for the tropospheric ozone production (Schwantes et al., 2020).

Field measurements indicate that there is a lack of understanding of the radical chemistry connected to the photooxidation of monoterpenes (e.g. Peeters et al., 2001; Capouet et al., 2004, and references therein). Hydroperoxy radical ($HO_2$) concentrations measured in environments dominated by monoterpenes are not well-understood by model calculations, for example in a campaign performed in the foothills of the Rocky Mountains (Kim et al., 2013) and in a boreal forest in Finland (Hens et al., 2014). In these campaigns, modeled $HO_2$ concentrations were lower by a factor of 2 and 2.5, respectively, compared to measurements. Also chamber studies, investigating the oxidation of $\alpha$- and $\beta$-pinene gave higher $HO_2$ concentrations than model calculations suggest (Kaminski et al., 2017; Rolletter et al., 2019).

Among the monoterpenes, $\alpha$-pinene is the most abundant one (Guenther et al., 2012). Pinonaldehyde is one of the first-generation oxidation products. In the $\alpha$-pinene photooxidation, initially 3 different peroxy radicals ($RO_2$) are formed and 2 of them eventually form pinonaldehyde. Several field campaigns reported pinonaldehyde concentrations measured on filter samples (Kavouras et al., 1999; Alves et al., 2002; Boy et al., 2004; Cahill et al., 2006; Rissanen et al., 2006; Herckes et al., 2006; Plewka et al., 2006) or in the analysis of rain and snow samples (Satsumabayashi et al., 2001). Only few studies reported ambient gas phase concentrations of pinonaldehyde. Field studies in the San Bernardino National Forest, California (Yu et al., 1999) and the German Fichtelgebirge (Müller et al., 2006) reported pinonaldehyde concentrations in the gas phase of approximately 0.15 ppbv. Another study by Cahill et al. (2006) conducted in the Sierra Nevada Mountains of California measured 0.05 to 0.30 ppbv gaseous pinonaldehyde. Pinonaldehyde yields from the photochemical degradation of $\alpha$-pinene with OH were measured in laboratory experiments ranging from 6 to 87 % (Larsen et al., 2001; Noziére et al., 1999). The high variability could be due to different chemical conditions with partly high reactant concentrations used in these experiments. A recent study in the atmospheric simulation chamber SAPHIR used ambient reactant concentrations ([$\alpha$-pinene] $\leq 3.8$ ppbv, [NO] $< 120$ pptv, $(310 \pm 5)$ K) and reported a low yield of 5 % (Rolletter et al., 2019). As currently implemented in the Master Chemical Mechanism (MCM, 2017; Jenkin et al., 1997; Saunders et al., 2003) pinonaldehyde is formed with a total yield of 84 %. In contrast, a theory based study by Vereecken et al. (2007) suggested a different branching ratio of initial $RO_2$ and additional reaction channels which lead in total to pinonaldehyde yields of 60 % for low atmospheric NO conditions ($\leq 1$ ppbv NO). Our previous study (Rolletter et al., 2019) showed that further adjustment of the initial $RO_2$ branching ratio in a mecha-

nism based on Vereecken et al. (2007) was necessary to explain the low measured pinonaldehyde yield of 5 % for conditions similar to the experiments discussed here. A similar change in $RO_2$ branching ratios was found in an experimental study by Xu et al. (2019).

The photochemistry of pinonaldehyde has been investigated in only few experimental studies (Noziére et al., 1999; Jaoui and Kamens, 2003; Capouet et al., 2004) in addition to few theoretical studies (Glasius et al., 1997; Vereecken and Peeters, 2002; Fantechi et al., 2002; Dash and Rajakumar, 2012). The pinonaldehyde degradation during daytime is controlled by photolysis and the reaction with OH radicals resulting in an atmospheric lifetime on the order of a few hours due to its fast reaction with OH (Atkinson et al., 2004). The lifetime of pinonaldehyde with respect to photodissociation strongly depends on the season and the latitude, but can be comparable to the reaction with OH (Hallquist et al., 1997). Main oxidation products (Fig. 1) of the photolysis and OH reactions are norpinonaldehyde (NORPINAL; names are taken from the MCM), pinonic acid (PINONIC), perpinonic acid (PERPINONIC), formaldehyde (HCHO), and acetone. So far, there has been only one study by Noziére et al. (1999) reporting yields for the reaction with OH for HCHO and acetone of $(152 \pm 56)$ % and $(15 \pm 7)$ %, respectively.

A simplified oxidation scheme of pinonaldehyde is shown in Fig. 1. As implemented in the MCM, the photolysis of pinonaldehyde (PINAL) leads to dissociation in HCO and an organic radical, both of which react with $O_2$ to form CO, $HO_2$ and the peroxy radical C96O2.

$$PINAL + h\nu + O_2 \quad \rightarrow \quad C96O2 + CO + HO_2 \tag{R1}$$

The oxidation of pinonaldehyde by OH is initiated by H-abstraction mainly at the aldehyde group forming primary $RO_2$. In the MCM, an acyl peroxy radical (C96CO3) and PINALO2 are formed.

$$PINAL + OH + O_2 \quad \rightarrow \quad C96CO3 \quad (yield: 0.77) \tag{R2}$$
$$PINAL + OH + O_2 \quad \rightarrow \quad PINALO2 \quad (yield: 0.23) \tag{R3}$$

In the presence of high nitric oxide (NO) concentrations, $RO_2$ react with NO and form either alkoxy radicals (RO) or organic nitrate compounds ($RONO_2$). In the reaction forming an alkoxy radical, NO is converted to $NO_2$ and $HO_2$ to OH, respectively. The alkoxy radical subsequently reacts with $O_2$ to form a carbonyl compound and $HO_2$, decomposes or isomerizes. At low NO concentrations, $RO_2$ reactions with $HO_2$ terminating the radical chain reactions and forming stable hydroxyperoxides (ROOH) gain in importance. Formed stable organic products of $RO_2 + NO$ and $RO_2 + HO_2$ reactions can further react with OH or photolyse.

The reaction of the most important $RO_2$ radical, C96CO3 with $HO_2$ can either directly form the same product (C96O2) as its reaction with NO or can form the stable products pinonic and perpinonic acid. Subsequently, pinonic and perpinonic acid are photolyzed or react with OH and thereby producing again the peroxy radical C96O2. Similarly, other $RO_2 + HO_2$ reactions not shown in Fig. 1 produce the same species which are produced by $RO_2 + NO$ reactions.

In the atmosphere, the peroxy radical C96O2 can react with nitric oxide (NO) to form the corresponding alkoxy radical C96O.

$$C96O2 + NO \quad \rightarrow \quad C96O + NO_2 \tag{R4}$$

C96O undergoes isomerization or decomposition reactions producing formaldehyde, acetone or 3,4-dioxopentanal (CO23C4CHO).

The main products of the subsequent chemistry of the other peroxy radical PINALO2, including multiple reactions with NO and decomposition reactions, are $HO_2$, a tri-carbonyl compound, acetone, and formaldehyde.

In contrast to the MCM, the theoretical study by Fantechi et al. (2002) predicts other decomposition reactions of products of the $RO_2 + NO$ reaction (Fig. 1). Whereas other theory based studies investigated the reaction rate of pinonaldehyde + OH (Glasius et al., 1997) and the probability of H-abstraction by OH at various sites of pinonaldehyde (Vereecken and Peeters, 2002; Dash and Rajakumar, 2012), Fantechi et al. (2002) analysed also the subsequent chemistry of the initially formed organic peroxy radicals. Instead of forming acetone and formaldehyde in the further degradation of C96O, Fantechi et al. (2002)

suggest that the the main fraction of C96O undergoes a series of isomerization reactions and reactions with NO forming 4-hydroxynorpinonaldehyde. In contrast to the MCM, the four-membered ring structure is retained in the formed products. A small fraction (~ 7%) directly forms norpinonaldehyde (NORPINAL) after H-abstraction by $O_2$. The modifications by Fantechi et al. (2002) are limited to $RO_2 + NO$ chemistry and no detailed analysis of $RO_2 + HO_2$ reactions was done in their work. Nevertheless, modifications suggested by Fantechi et al. (2002) also affect $RO_2 + HO_2$ chemistry, if products from these

reactions are the same as from the reaction of $RO_2 + NO$ as implemented in the MCM. In the model by Fantechi et al. (2002) some $RO_2 + NO$ reactions also lead to the formation of organic nitrates with yields between 14 % and 28 %, which lower the yield of formed $HO_2$ and carbonyl compounds.

Fantechi et al. (2002) proposed two additional relevant (yields $> 5$ %) peroxy radicals formed by the initial OH attack. Accordingly, the yields of C96CO3 and PINALO2 are changed from 77 % to 61 % and 23 % to 9 % compared to the MCM,

respectively. The first additional reaction channel with a branching ratio of 24 % leads to a formation of norpinonaldehyde. The second additional reaction channel with only a minor contribution of 6 % forms in the subsequent chemistry acetone, formaldehyde and a tri-carbonyl-hydroxy compound.

In this work, the photochemistry of pinonaldehyde was investigated under controlled, atmospheric conditions including atmospherically reactant concentrations and natural sunlight in the outdoor chamber SAPHIR (Simulation of Atmospheric

PHotochemistry In a large Reaction Chamber) at Forschungszentrum Jülich. This study focusses on (a) the determination of the pinonaldehyde photolysis frequency, (b) quantification of $HO_x$ ($= OH + HO_2$) radicals in the OH–oxidation of pinonalde-hyde, and (c) the determination of acetone and formaldehyde yields of both photolysis and OH–oxidation. Measurements of pinonaldehyde, degradation products and $HO_X$ radicals are compared to model calculations applying the Master Chemical Mechanism v3.3.1. Sensitivity model runs are performed including reaction pathways and yields suggested by the theoretical

study of Fantechi et al. (2002) and the impact of the proposed mechanism on concentrations of organic compounds and $HO_x$ radicals is analyzed.

## 2 Methods

### 2.1 Atmospheric simulation chamber SAPHIR

Details of the SAPHIR chamber can be found elsewhere (e.g. Rohrer et al., 2005; Bohn et al., 2005) and the chamber is only briefly described here. It is of cylindrical shape (18 m length, 5 m diameter, and 270 m$^3$ volume) and is made of a double wall FEP film. The FEP ensures inertness of the surfaces and minimizes wall effects. This outdoor chamber allows studying the photochemical reactions under natural sunlight, because the FEP film is transmissive for the entire solar spectrum. The chamber is equipped with a shutter system that can shield the chamber from sunlight.

Ultra-pure synthetic air used in the experiments is mixed from liquid nitrogen and oxygen (Linde, purity >99.9999 %). The pressure inside the chamber is slightly (30 Pa) above ambient pressure to prevent impurities from ambient air leaking into the chamber. Air that is consumed by instruments or is lost due to small leakages is replenished to keep a constant pressure. The replenishment flow is in the range of 9 to 12 m$^3$h$^{-1}$ leading to a dilution of trace gases of 3 to 4 % h$^{-1}$. Rapid mixing of air is ensured by the operation of two fans inside the chamber.

In the sunlit chamber, there are small sources for nitrous acid (HONO), formaldehyde (HCHO), and acetone. Their formation rates depend on the intensity of solar radiation, relative humidity and temperature (Rohrer et al., 2005). The photolysis of HONO is the major primary source for nitrogen oxides and OH. Approximately, 250 pptv h$^{-1}$ NO$_x$, 200 pptv h$^{-1}$ HCHO and 100 pptv h$^{-1}$ acetone were formed in the experiments in this study.

### 2.2 Instrumentation

An overview of used instruments and their 1 $\sigma$ accuracies and precision is given in Table 1.

OH radicals were measured by laser induced fluorescence (LIF). The instrument operated at the SAPHIR chamber has been described elsewhere (e.g. Holland et al., 2003; Fuchs et al., 2012). Chamber air is sampled through an inlet nozzle into a low-pressure measurement cell, where OH is excited by pulsed laser radiation at 308 nm. Its fluorescence is subsequently detected by gated photon counting. In addition, HO$_2$ radicals are indirectly detected after chemical conversion to OH by NO in another measurement cell (Fuchs et al., 2011). The instrument is calibrated by a radical source, in which OH and HO$_2$ are produced by water vapor photolysis at 185 nm (Fuchs et al., 2011).

Interferences can occur in the HO$_2$ measurements because of concurrent conversion of specific RO$_2$ radicals which produce OH in the reaction with NO on a similar time scale as HO$_2$ does (Fuchs et al., 2011; Whalley et al., 2013; Lew et al., 2018). Because the conversion of RO$_2$ requires at least two reaction steps with NO in contrast to HO$_2$ for which only one reaction step leads to the formation of OH, this interference can be minimized, if the conversion efficiency of HO$_2$ is below 10 %. This can be achieved by adjusting the NO concentration in the HO$_2$ cell. This was done in this study, so that it can be assumed that potential interferences in the HO$_2$ detection were negligible.

In addition, OH was measured by differential optical absorption spectrometry (DOAS; Dorn et al., 1995b) which is an absolute measurement technique. Measured OH concentrations of both instruments agreed on average within 15 % similar to results in previous experiments (Fuchs et al., 2012).

Organic compounds were measured by proton transfer reaction time-of-flight mass spectrometry (PTR-TOF-MS; Lindinger et al., 1998; Jordan et al., 2009) and by gas chromatography that was coupled to a flame ionization detector (GC-FID). The PTR-TOF-MS was calibrated for pinonaldehyde using a diffusion source (Gautrois and Koppmann, 1999) and for acetone using a gas standard. An internal standard (tetrachloroethylene) was added to the GC-FID sample flow to to monitor changes in the sensitivity of the calibration with calibration standard (Apel 8). Formaldehyde was measured by a Hantzsch instrument and by DOAS. The Hantzsch instrument was calibrated using liquid HCHO standards.

Photolysis frequencies (j) of $NO_2$, HONO, $O_3$, and pinonaldehyde were calculated from actinic flux density spectra that are derived from measurements of total and diffuse spectral actinic flux densities outside the chamber. From these measurements direct sun contributions are calculated. The direct and diffuse actinic flux densities are used as input for a model which calculates mean chamber spectra by taking into account the time-dependent effects of shadings of the chamber steel frame and the transmittance of the Teflon film which is > 0.8 in the complete solar spectral range (Bohn and Zilken, 2005). Mean photolysis frequencies are then calculated by the following equation:

$$j = \int \sigma(\lambda)\phi(\lambda)F_\lambda(\lambda)\mathrm{d}\lambda \tag{1}$$

with $\sigma$ the absorption cross section, $\phi$ the quantum yield, and F the actinic flux. The absorption cross sections and quantum yields of $NO_2$, HONO, and $O_3$ used for the calculations of photolysis frequencies are taken from the literature (Mérienne et al., 1995; Troe, 2000; Stutz et al., 2000; Daumont et al., 1992; Matsumi et al., 2002). The method is regularly evaluated by dedicated experiments using the chamber as a chemical actinometer (Bohn et al., 2005). The pinonaldehyde photolysis frequency is discussed in more detail in Section 4.1.

NO and $NO_2$ were measured by chemiluminescence (Eco Physics), water vapor mixing ratios by a cavity ring-down instrument (Picarro) and ozone ($O_3$) by an UV absorption instrument (Ansyco).

## 2.3 Experimental procedure

Before the experiment, the chamber was flushed with dry, synthetic air to dilute trace gases from previous experiments below the detection limits of the instruments. Twenty ppm of $CO_2$ was injected as a dilution tracer in the beginning of each experiment. Water from a Milli-Q-device was boiled and flushed into the chamber together with a high flow of synthetic air ($150\,\mathrm{m^3h^{-1}}$). The chamber was only humidified in the beginning of an experiment reaching initial water vapor concentrations of about 2 % which decreased over the course of an experiment due to the dilution with dry synthetic air.

Pinonaldehyde (Orgentis chemicals, 98.2 %) was heated in a glass vial and the vapor was flushed together with a small flow of dry nitrogen into the chamber using a short Teflon tube. After the injection had been stopped, the sample line was removed to avoid further evaporation of pinonaldehyde from the injection system into the chamber. The initial pinonaldehyde concentrations were 6.5 and 16.5 ppbv in the experiments on 17 July and 18 July 2014, respectively. The chamber roof was opened after a stable pinonaldehyde concentration was observed by the PTR-TOF-MS instrument.

In one experiment (18 July 2014), 2500 ppbv of cyclohexane was additionally injected after the humidification. Cyclohexane served as scavenger for OH in this experiment in order to study the pinonaldehyde photolysis independently from its reaction with OH.

In the other experiment (17 July 2014), 70 ppbv of ozone produced from a silent discharge ozonizer (O3Onia) was injected after humidification, so that a low NO concentration (maximum 80 pptv) was obtained during the experiment. In this experiment, approximately 20 % of reacted pinonaldehyde was photolyzed and the remaining 80 % reacted with OH. The reaction of pinonaldehyde with ozone is very slow (k = (8.9 ± 1.4) x $10^{-20}$cm$^3$s$^{-1}$, Glasius et al. (1997)) and therefore ozonolysis reactions did not play a role in the experiments here.

## 2.4 Model calculations

The Master Chemical Mechanism (MCM) in its latest version 3.3.1 was applied as base model for box model calculations (MCM, 2017; Jenkin et al., 1997; Saunders et al., 2003). FACSIMILE was used as solver for differential equations in the model calculations.

In order to account for chamber effects, the following modifications were added to the MCM model. Dilution to all trace gases was applied. The dilution rate was calculated from the monitored replenishment flow rate that was consistent with the dilution of the $CO_2$ tracer. Small chamber sources of formaldehyde and acetone were parameterized based on reference experiments as described in Rohrer et al. (2005); Karl et al. (2006); Kaminski et al. (2017).

Temperature, pressure, $H_2O$, NO, $NO_2$, HONO, and $O_3$ concentrations were constrained to measured values. These constrains were used because chamber NO sources cannot be modeled accurately and therefore could lead to wrong conclusions in the analysis of turnover rates of radicals.

Photolysis frequencies, which were not measured, were calculated for clear sky conditions by the parameterization included in the MCM. They were scaled to take cloud cover and the transmission of the Teflon film into account by the ratio of measured to modeled photolysis frequency of $NO_2$. This applies also for the photolysis frequency of pinonaldehyde in the base model case (MCM). All other model runs used the experimentally derived pinonaldehyde photolysis frequency (see Section 4.1).

The injection of pinonaldehyde was modeled as a source only present during the time period of injection to match the pinonaldehyde increase as measured by PTR-TOF-MS.

In a sensitivity study, the pinonaldehyde oxidation scheme developed by Fantechi et al. (2002) was implemented. In this mechanism, additional reaction pathways of the pinonaldehyde reaction with OH that are not included in the MCM are suggested. Only the four reaction pathways with significant yields (> 5 %) were implemented for the sensitivity model run (Fig. 1). In addition, the fate of the most abundant $RO_2$ formed from the reaction of pinonaldehyde with OH or photolysis is different in this model compared to the MCM mechanism (see above). Reaction rate constants and branching ratios were used as described in Fantechi et al. (2002). Simple rate coefficients such as KDEC (= 1 x $10^6$) and KRO2NO (= 2.7 x $10^{-12}$ exp(360/T)) from the MCM were used, when no specific reaction rate constants were mentioned. To account for possible $RO_2$ + $HO_2$ reactions, a reaction scheme based on the reaction C97O2 + $HO_2$ was added for all newly introduced $RO_2$ species not included in the MCM. $RO_2$ + $HO_2$ reactions form a corresponding hydroxyperoxide (ROOH) that can either react with OH to regenerate the $RO_2$

or photolyse to form the corresponding alkoxy radical (RO). Modifications by Fantechi et al. (2002) describe the chemistry from the OH attack on pinonaldehyde until the formation of stable products 4-hydroxynorpinonaldehyde, norpinonaldehyde, CO23C4CHO, and C818CO. The chemistry of the products norpinonaldehyde, CO23C4CHO and C818CO was treated as described in the MCM. For 4-hydroxynorpinonaldehyde, no follow-up chemistry was considered.

5    Both MCM and Fantechi et al. (2002) mechanism were tested under different conditions. An overview of used model calulations is given in Table 2.

## 3   Results

Figures 2 and 3 show the concentration time series of pinonaldehyde and other measured trace gases together with MCM model results for the two experiments with and without OH scavenger, respectively.

10   Weather conditions were similar for both experiment days. The temperature inside the chamber increased over the course of each experiment from 305 to 315 K. The solar zenith angle at noon was 30 ° and maximum measured pinonaldehyde photolysis frequencies were approximately $3.5 \times 10^{-5}$ s$^{-1}$ in both experiments.

In both experiments, the measured pinonaldehyde decay was significantly faster than predicted by the model once the chamber roof was opened. The predicted pinonaldehyde consumption rate was slower by 50 % and 25 % in the pure photolysis case and in the OH oxidation case, respectively. Potential wall loss of pinonaldehyde in the chamber was tested in separate experiments, in which pinonaldehyde was injected into the dark chamber. The loss of pinonaldehyde in this case is consistent with the dilution calculated from the replenishment flow.

Also the two measured major organic products, acetone and formaldehyde, are not well reproduced by the model calculations. In the experiment with the OH scavenger, in which pinonaldehyde is removed by photolysis only, the modeled concentrations of formaldehyde and acetone are 60 % and 70 % higher compared to the measurements, although the photolysis frequency of pinonaldehyde is obviously underestimated. In contrast, in the experiment in which pinonaldehyde was photolyzed and oxidized by OH modeled acetone concentrations are underestimated by 17 % compared to measurements. Modeled formaldehyde concentrations are 6 % smaller than measurements, which is within the measurement uncertainty. The smaller modeled acetone yield cannot be explained only by the lower consumption of pinonaldehyde in the model. A sensitivity run which reproduces the pinonaldehyde consumption also underestimates the acetone concentrations.

In the experiment without scavenger, OH and HO$_2$ are both considerably underpredicted by the MCM. While OH shows an increasing discrepancy of up to a factor of 2, modeled HO$_2$ is a factor of 2–4 lower than measured. In the experiment with OH scavenger, measured and modeled HO$_2$ concentrations agree within the stated uncertainty during the first part of the experiment, but the model underestimates the HO$_2$ by 50 % in the last hour.

30   Measured time series of acetone and formaldehyde are used to determine the total yields of acetone and formaldehyde from the pinonaldehyde oxidation following the procedure described by Galloway et al. (2011), Kaminski et al. (2017), and Rolletter et al. (2019). In this approach, the measured time series of trace gases are corrected for loss and production that are not directly related to the chemical oxidation scheme of pinonaldehyde. This includes dilution of trace gases in the chamber, loss

of formaldehyde due to photolysis and small production of formaldehyde and acetone in the sunlit chamber that is independent from the pinonaldehyde chemistry. If also OH is present, additional corrections for the losses of formaldehyde and acetone due to their reactions with OH are applied. The corrected time series can then be used to calculate the ratio of a formed organic product and the consumed pinonaldehyde to derive the yield of the organic compound that is connected to the chemical degradation of pinonaldehyde. The main uncertainty in the calculated yields is caused by the uncertainty of the strength of the small chamber sources which has to be taken from characterization experiments that were performed before/after the experiments with pinonaldehyde. Sensitivity studies in which the source strengths are systematically varied show that $1\sigma$ uncertainties of yields are 0.2 and 0.1 for formaldehyde and acetone, respectively.

Figure 4 shows the result of the yield calculations. The formaldehyde yield is $(0.1 \pm 0.2)$ in the beginning and increases to approximately $(0.18 \pm 0.2)$ in the end of the experiment. The acetone yield of the photolysis is zero within the uncertainty of 0.1.

The yields of acetone and formaldehyde are also calculated from the measured time series in the experiment when pinonaldehyde was oxidized by OH as described for the experiment with OH scavenger. Results are shown in Fig. 5. The yields of both species increase over the course of the experiment. The formaldehyde yield increases from approximately $(0.15 \pm 0.2)$ to values higher than $(0.45 \pm 0.2)$ and the acetone yield from approximately $(0.2 \pm 0.1)$ to $(0.3 \pm 0.1)$.

## 4 Discussion

### 4.1 Pinonaldehyde photolysis

In the presence of an OH scavenger, the pinonaldehyde decay observed in the chamber could be due to photolysis, wall loss and dilution. As was shown by additional experiments in the dark chamber, wall loss is negligible on the time scale of a few hours. The effect of dilution was quantified from the measured dilution flow rate and the chamber volume and agreed with the results from the tracer ($CO_2$) measurements within 3 %. Without OH, the chemical degradation of pinonaldehyde depends only on the photolysis frequency of pinonaldehyde. In MCM, photolysis frequencies are generally calculated as a function of solar zenith angle $\chi$ using three parameters l, m, and n.

$$J = l \cdot (\cos(\chi))^m \cdot e^{(-n \cdot \sec(\chi))} \qquad (2)$$

For pinonaldehyde, MCM is using the parameters ($l = 2.792 \times 10^{-5} s^{-1}$, m = 0.805, n = 0.338; valid for clear sky conditions) for the photolysis of n-butanal. The parameterization is based on the absorption spectrum measured for n-butanal at 298 K by Martinez et al. (1992) and the quantum yield (0.21) for its dissociation to n-$C_3H_7$ + HCO at 298 K and 1 atm by Tadić et al. (2001). For the evaluation of the chamber experiments, the clear sky values from the parameterization (Equation 1) are corrected for the influence of cloud cover and chamber transmission by multiplying the clear sky value by the ratio of the measured to parametrized photolysis frequency of $NO_2$. Following this procedure, the simulated decay of pinonaldehyde in the base model (MCM) is considerably slower than the observed decay (Fig. 6).

As an alternative, the photolysis frequency of pinonaldehyde is calculated using the measured spectrally-resolved solar actinic flux (Section 2.2) and the pinonaldehyde absorption spectrum (280-340 nm) measured at 300 K by Hallquist et al. (1997). Good agreement between the observed and simulated (MCM_a) pinonaldehyde decay is achieved, if an effective quantum yield of 0.9 is assumed (Fig. 6, upper panel). Here, the mean decay rate of pinonaldehyde between 10:00 and 15:00 (UTC)

is $3.9 \times 10^{-5}\,\mathrm{s}^{-1}$, of which $2.9 \times 10^{-5}\,\mathrm{s}^{-1}$ is caused by photolysis and $1.0 \times 10^{-5}\,\mathrm{s}^{-1}$ by dilution. The experimentally derived photolysis frequencies are approximately a factor of 3 to 3.5 higher than the values from the parameterization in the MCM. The experimental error (20 %) of the effective quantum yield (0.9) is mainly determined by the uncertainties of the absorption spectrum (< 5 %; Hallquist et al. (1997)) and the actinic flux measurement (10 %). Error contributions from wall loss and dilution are small (< 5 %).

The applied absorption cross sections by Hallquist et al. (1997) are the only published measurements and are recommended by IUPAC (Atkinson et al., 2006). The effective, wavelength-independent quantum yield determined in this work is significantly higher than in two other chamber studies, which report values of $0.14 \pm 0.03$ (Moortgat et al., 2002) and 0.4 (Jaoui and Kamens, 2003) for photolysis with natural sunlight. No recommendation for the quantum yield is given by IUPAC (Atkinson et al., 2006). Both chamber studies applied the absorption spectrum from Hallquist et al. (1997) for the calculation of photolysis frequencies. Jaoui and Kamens (2003) measured the solar radiation by a UV-Eppley radiometer. The broadband instrument measures the spectrally integrated solar irradiance (spatially cosine-weighted photon-flux density) from 300 nm to 400 nm. The non-trivial conversion to a actinic flux spectrum (spatially isotropically weighted photon-flux density) between 300 nm and 340 nm needed for the evaluation of the pinonaldehyde photoloysis frequencies has not been documented by the authors. The conversion requires knowledge of the spatial distribution of the incident solar radiation, which is a function of solar zenith angle, wavelength, atmospheric aerosol and clouds (Hofzumahaus, 2006). Furthermore, the wavelength range of pinonaldehyde photolysis (< 340 nm) strongly depends on the total atmospheric ozone column, while UV Eppley measurements (300-400 nm) are only weakly dependent on total ozone. Considerable errors may therefore be connected to the conversion of Eppley data to photolysis frequencies for pinonaldehyde.

In the study by Moortgat et al. (2002), solar actinic flux was directly measured by a spectroradiometer with good accuracy like in the present work. However, similar to Jaoui and Kamens (2003), Moortgat et al. (2002) had to apply large corrections for wall losses and dilution, each of which were of the same magnitude as the photolysis rate. The large difference by a factor of 7 in comparison to the present work is likely not explained by systematic errors of the correction.

The pinonaldehyde photolysis is faster than n-butanal because of its two carbonyl functions. This might be valid for other bi-carbonyl compounds that have non-conjugated carbonyl functions, so that the use of the n-butanal photolysis frequency could systematically underestimate the photolysis frequencies of these compounds. However, the high quantum yield close to unity could also be a specific property of pinonaldehyde that might not apply for the photolysis of other bi-carbonyl species.

Figure 4 shows the time series of measured acetone and formaldehyde concentrations together with results from model calculations applying the MCM. In contrast to the base case model (MCM, Fig. 2), measured photolysis frequencies are used (MCM_a). As a consequence of the the higher photolysis frequencies, the consumption of pinonaldehyde leads to even larger

productions of formaldehyde and acetone (approximately three times higher than measured values) compared to the base case model.

For conditions of this experiment, 80 % of $RO_2$ radicals formed in the photolysis reaction of pinonaldehyde are reacting with NO and only 20 % are reacting with $HO_2$, so that carbonyl compounds are expected to be the main organic products (Fig. 1). Acetone measured in this experiment is solely formed by the chamber source. Initial acetone yield values are negative because of the high uncertainty in the corrections that are applied in the yield calculation. In the beginning of an experiment, only small amounts of products are formed which leads to a large uncertainty, so that negative values are not significant. It has to be stressed here again that the parametrization of the chamber source is the main uncertainty in the yield calculation. If the chamber source was overestimated, the constant measured acetone yield could also include a small contribution from the pinonaldehyde photolysis.

In a sensitivity model run (FAN_a), the pinonaldehyde oxidation scheme suggested by Fantechi et al. (2002) is tested for the experiment with OH scavenger. While the initial photodissociation step and the reaction of C96O2 with NO are the same, the following decomposition of C96O yields considerably different organic products. No acetone and less formaldehyde are produced together with mainly 4-hydroxynorpinonaldehyde and norpinonaldehyde that were not measured in these experiments. Agreement between modeled and measured acetone is achieved within the accuracy of measurements of 10 %. Also modeled formaldehyde concentrations are only 20 % lower than measured values. Thus, the model description of acetone and formaldehyde products is greatly improved by the use of the Fantechi et al. (2002) mechanism compared to the MCM.

To our knowledge, there is only one other study by Jaoui and Kamens (2003), which investigated the product yields of pinonaldehyde photolysis. Products were measured by gas-chromatography in that chamber study in the presence and absence of an OH scavenger. The measured norpinonaldehyde yield agrees within the stated uncertainty with the yield proposed by Fantechi et al. (2002). Formaldehyde and acetone yields were not measured by Jaoui and Kamens (2003).

Using the MCM with the measured photolysis frequency (MCM_a) leads to an increase in modeled $HO_2$ of about 25 % (Fig. 6, lower panel) which can be explained by the higher amount of consumed pinonaldehyde that is formed at the end of the radical reaction chain together with 3,4-dioxopentanal (CO23C4CHO). Unfortunately, between 11:30 and 14:00 experimental problems occurred in the $HO_2$ measurements. Neither NO measurements nor photolysis frequencies showed any features that could explain the decrease in the $HO_2$ concentration. The exact reason of the $HO_2$ variations remains unclear and the uncertainty of $HO_2$ measurements is likely higher (50 %) for this period. Implementing the modifications by Fantechi et al. (2002) results in a $HO_2$ concentration time profile (FAN_a) that is different from both model runs done with the MCM mechanism. In the FAN_a model run $HO_2$ is formed more rapidly compared to the MCM and concentrations decrease towards the end of the experiment. The rate determining step in radical chain reactions is the reaction of $RO_2$ with NO forming an alkoxy radical and $NO_2$. In the MCM, there are 3 $RO_2 + NO$ reactions before the radical chain is terminated and the stable product CO23C4CHO is formed together with $HO_2$. In contrast, in the Fantechi et al. (2002) mechanism only one $RO_2 + NO$ reaction occurs before the stable products are formed and $HO_2$ is regenerated. In addition, no subsequent chemistry of the formed product 4-hydroxynorpinonaldehyde is included in the model which would produce additional $HO_2$, especially in later stages of the experiment.

## 4.2   Photooxidation by OH

In the photooxidation experiment without OH scavenger (Fig. 3), a much faster decay of pinonaldehyde is observed compared to the case with OH scavenger (Fig. 2). Without scavenger, 30 % of the decay in the beginning of the experiment is explained by photolysis and 5 % by dilution. The remaining 65 % are due to the removal of pinonaldehyde by OH. However, OH concentrations rise over the course of the experiment and the pinonaldehyde fraction reacting with OH increase to up to 80 %. Here and in the following analysis, experimental photolysis frequencies are used, which assume a quantum yield of 0.9 (see Section 4.1). The photolysis and the OH reaction of pinonaldehyde lead to a mix of peroxy radicals (Fig. 1), which react mainly with $HO_2$ or NO. During the experiment, the NO mixing ratio increases from about 10 pptv to 80 pptv. Accordingly, the fraction of $RO_2$ radicals reacting with NO increases from 20 % to 55 %, while the $RO_2$ fraction reacting with $HO_2$ decreases.

Applying the experimental pinonaldehyde photolysis frequency to the model (MCM_a) improves the simulation of the pinonaldehyde decay compared to the base model run (MCM), which uses a slower parameterized photolysis frequency. However, even with the faster photolysis rate, the consumption of pinonaldehyde is underestimated (Fig. 7, upper panel). At the end of the experiment (15:00), the remaining modeled pinonaldehyde concentration is a factor of 2 larger than the measured value. This is due to the lower OH concentration in the model compared to measurements. If modeled OH concentrations are increased to match measured values, which can be achieved by constraining $HO_2$ in the model to measurements (MCM_b), modeled and measured pinonaldehyde concentrations agree within 10 % (Fig. 7). This demonstrates that missing OH production in the model is most likely due to the underestimation of $HO_2$, which forms OH by reaction with NO. This suggests that there is a $HO_2$ source missing in the model. In a sensitivity run (not shown here) artificial $HO_2$ sources were added to the model to quantify the required $HO_2$ source strength. The missing $HO_2$ source is increasing over the course of the experiment. A source strength between $0.8\,\mathrm{ppbv\,h^{-1}}$ and $1.5\,\mathrm{ppbv\,h^{-1}}$ would be required to explain observations. The consumption of pinonaldehyde during the experiment is up to $2.5\,\mathrm{ppbv\,h^{-1}}$, so that a comparably large $HO_2$ source would be required.

Implementation of the mechanism by Fantechi et al. (2002) does not improve the model-measurement agreement of OH and $HO_2$ (Fig. 8) compared to the MCM. As discussed above, the mechanism by Fantechi et al. (2002) is able to describe $HO_2$ concentrations as well as product formation of acetone and HCHO, if pinonaldehyde is only consumed by photolysis. In contrast, reaction pathways that are connected to the reaction of pinonaldehyde + OH are not correctly described.

Modeled $HO_2$ concentrations could be affected by the use of general reaction rate constants for reactions of $RO_2$ with $HO_2$ and NO (KRO2HO2 and KRO2NO), respectively. This might be an oversimplification for highly functionalized compounds. A sensitivity test (see Supplement) with an enhanced reaction rate for $RO_2 + NO$ reactions of $2\times$ KRO2NO in the modified mechanism by Fantechi et al. (2002) was performed. As a result, the fraction of $RO_2$ reacting with NO instead of $HO_2$ is increased. This leads to an enhanced $HO_2$ concentration of approximately 50 % compared to the model run FAN_a. However, $HO_x$ concentrations are again underestimated compared to measurements and the sensitivity run cannot reproduce the $HO_2$ concentration time behaviour from observations.

Besides reactions with NO, new types of $RO_2$ reactions have been recognized in the last decade that can produce $HO_x$. These processes include unimolecular autoxidation reactions of $RO_2$ (e.g. Crounse et al., 2013) and reactions of $RO_2$ with $HO_2$

yielding OH (e.g. Hasson et al., 2004). The new types of reactions become potentially important, when NO concentrations are below 1 ppbv. These reactions are especially favoured when the $RO_2$ molecule contains functional groups, like carbonyl groups as in the case of pinonaldehyde peroxy radicals. For example, acetyl- and acetonyl-$RO_2$ can react with $HO_2$ forming alkoxy radicals and OH instead of terminating the radical reaction chain forming hydroxyperoxides (Hasson et al., 2004; Dillon and

Crowley, 2008). Other examples are reactions already included in the MCM, such as the photooxidation of methyl vinyl ketone (Praske et al., 2015; Fuchs et al., 2018). For pinonaldehyde, there have been no comparable reactions suggested so far, although pinonaldehyde and its degradation products have at least 2 carbonyl functions. Because 80 to 45 % of $RO_2$ reacts with $HO_2$ for conditions of the experiment here, such reaction pathways have the potential to impact the OH production rate. A sensitivity test (not shown here) shows that $RO_2 + HO_2$ reactions of $RO_2$ included in the mechanism with a rate of $10\times$ KRO2HO2

can reproduce measured OH concentrations. Nevertheless, the enhanced $HO_2$ consumption increases the model-measurement discrepancy of $HO_2$ even more.

Reported autoxidation reactions of $RO_2$, which produce $HO_x$ without NO, involve isomerization and decomposition of organic peroxy radicals. These reactions play a role, for example, in the photooxidation of isoprene (e.g. Fuchs et al., 2013; Peeters et al., 2014; Novelli et al., 2020) and methacrolein (Crounse et al., 2012; Fuchs et al., 2014), where H-shift reactions

in $RO_2$ species lead to decomposition into a radical (OH or $HO_2$) and a carbonyl compound. In general, rate coefficients of H-shift reactions are strongly enhanced by the presence of functional groups such as carbonyl groups, and can reach values in the order of 0.1 s$^{-1}$ at room temperature (Crounse et al., 2013; Otkjær et al., 2018). Depending on the specific $RO_2$ structure and its functional groups, either OH or $HO_2$ can be formed. The presence of hydroxyl or hydroperoxy groups in carbonyl peroxy radicals, for example, favours the elimination of $HO_2$.

In the chemical degradation of pinonaldehyde to its first-generation products (Fig. 1), a large number of multifunctional peroxy and alkoxy radicals are formed as intermediates. Thus, there is potential for additional $HO_2$ formation by unimolecular reactions. This possibility is explored in a model sensitivity run for the four oxidation branches I - IV (Fantechi et al., 2002), which follow OH addition to pinonaldehyde. The model run S1 (see Supplement) assumes that each of the initially formed peroxy radicals (C96CO3, FAN_D1, PINALO2, and FAN_G1) is eventually converted to $HO_2$ with a rate coefficient of 0.1 s$^{-1}$.

However, only FAN_D1, PINALO2, and FAN_G1 have an aldehyde group with a hydrogen that can be easily abstracted (see Supplementary material). The model run (Fig. 8) shows a considerable enhancement of the $HO_2$ concentration level in the first period of the experiment compared to model runs MCM_a and FAN_a, leading to good agreement between modeled and measured OH. However, the temporal trend of the modeled $HO_2$ is not well described. While the observed $HO_2$ shows a steady increase from the beginning to the end of the experiment, the simulation S1 shows a continuous decrease which follows the

concentration of the short-lived $RO_2$ radicals. The opposite temporal trend suggests that additional $HO_2$ formation by a fast process in the oxidation branches I - IV is not a likely explanation. It indicates that the additionally required $HO_2$ is slowly built up, probably from stable products of the pinonaldehyde oxidation.

One such possibility would be the photolysis of first-generation products. This idea is tested in model run S2 (Table 2, Fig. 8). All products of the pinonaldehyde photooxidation have either two or three carbonyl groups and therefore are likely to undergo

photolysis. CO23C4CHO and C818CO even have conjugated carbonyl functions similar to glyoxal, which photolysis is up to

two times faster than pinonaldehyde. However, using the photolysis frequency of glyoxal as an upper limit for the photolysis frequency of the products formed here, is not sufficient to significantly improve the $HO_2$ model-measurement agreement. Only if a strongly enhanced photolysis frequency equivalent to $0.2 \times j_{NO2}$ is applied, modeled $HO_2$ comes close to the observed values. In this case, the temporal trend of the simulation is similar to the observed time behaviour of $HO_2$ and also OH is reasonably well reproduced. This supports the hypothesis that the additional $HO_2$ is slowly formed from stable oxidation products. However, the value for the assumed photolysis frequency, which is 200 times larger than of pinonaldehyde, appears unrealistically high.

Another possibility is that the fast H-shift isomerization of $RO_2$ radicals (see Supplement) leads to the formation of peroxy acids with additional carbonyl functions in high yields. As discussed above, these bi-functional compounds could photolyse faster than currently implemented in the mechanism. A sensitivity test (S1_mod_hv, see Supplement) was performed that includes isomerization of $RO_2$ with a –HCO group. Products are assumed to photolyse with a photolysis rate that is 2 times higher than that of glyoxal. Implementation of these reactions leads to $HO_2$ concentrations that are increased by up to 60 % compared to the sensitivity run that includes only isomerization reactions. Calculated $HO_2$ concentrations underestimate measurements by factor of 2. The sensitivity test reproduces measured OH concentrations within the measurement uncertainty.

No subsequent chemistry of 4-hydroxynorpinonaldehyde is included in the mechanism so far. In the experiment here 4-hydroxynorpinonaldehyde is formed with an overall yield of approximately 25 %. 4-hydroxynorpinonaldehyde is highly functionalized and $RO_2$ radicals formed in its degradation could undergo fast isomerization reactions. For a sensitivity run (S3, see Supplement) a mechanism was deduced with the structure–activity relationship (SAR; Kwok and Atkinson, 1995; Vereecken and Peeters, 2009; Vereecken and Nozière, 2020) method. However, the impact of the tested 4-hydroxynorpinonaldehyde degradation scheme on the $HO_2$ formation was small ($\leq 10$ %) compared to the modified mechanism by Fantechi et al. (2002). Unfortunately, no measurements of stable oxidation products other than acetone and HCHO were available. Without further product measurements the whole analysis discussed here relies on product distribution prescribed by the models. Further experiments that measure oxidation products and yields could help to better constrain branching ratios in degradation mechanisms. In addition, theoretical studies could investigate subsequent degradation schemes of major products in more detail.

The continuous increase in the acetone and formaldehyde yields during the experiment (Fig. 5) indicates that both species are not only formed from the first reaction step of pinonaldehyde with OH, but also from further oxidation of organic products.

The base model (Fig. 3, MCM) underestimates the pinonaldehyde consumption, but shows a good model-measurement agreement with formaldehyde and acetone within the measurement uncertainty. In contrast, the model, which uses measured pinonaldehyde photolysis frequencies and is constrained to measured $HO_2$, produces up to 30 % less acetone and formaldehyde than measured (Fig. 5, MCM_b). The discrepancies are increasing fast during the first 2 hours of the experiment, when pinonaldehyde is the most important reaction partner for OH, and slows down, when oxidation products gain importance at later times of the experiment. The elevated $HO_2$ concentrations change the product distribution compared to the base case with less formed formaldehyde and acetone, because $RO_2 + HO_2$ reactions producing hydroxyperoxides become more important compared to the $RO_2 + NO$ pathway. In the chemical model, acetone and formaldehyde of this reaction pathway are formed by the slow photolysis (10 times slower than the photolysis of pinonaldehyde) of pinonic acid and perpinonic acid that are pro-

duced in the subsequent chemistry of hydroxyperoxides. Therefore, acetone and formaldehyde yields are smaller in the MCM model run, if $HO_2$ concentrations are correctly described compared to the base case MCM model, when $HO_2$ is significantly underestimated.

Implementation of the mechanism by Fantechi et al. (2002) with also $HO_2$ concentrations and pinonaldehyde photolysis frequency constrained to measurements (FAN_b) makes the model-measurement agreement for acetone and formaldehyde worse. Acetone and formaldehyde yields are lowered and 4-hydroxynorpinonaldehyde and norpinonaldehyde are produced instead. Acetone and formaldehyde time series agree for the photolysis experiment, when the mechanism by Fantechi et al. (2002) is applied. Similarly, the majority of consumed pinonaldehyde (approximately 65 %) forms the peroxy radical C96O2 either by photolysis or reaction pathway I (Fig. 1), when also OH is present. Therefore, it can be assumed that the C96O2 + NO reaction channel is not responsible for the underprediction of acetone and formaldehyde at least for the early times of the experiments, when contributions from OH reactions of product species are small. However, because $RO_2 + HO_2$ reactions are more important in the experiment with OH oxidation (see above), additional $HO_2$ production from this reaction pathway has the potential to serve as an explanation for the observed disccrepancies. In addition, minor pathways could produce additional formaldehyde and acetone to explain the model-measurement discrepancy right after the start of the pinonaldehyde oxidation. At later times of the experiment, additional production of acetone and formaldehyde from the further degradation of oxidation products need to be assumed to close the gap. For example, this could be due to a reaction channel of the alkoxy radical R$'$O8 that does not produce norpinonaldehyde, but acetone and formaldehyde instead. However, the exact chemical mechanism that is responsible for the additional acetone and formaldehyde cannot be determined from measurements in these experiments.

Presently, there is only one work of Noziére et al. (1999) where acetone and formaldehyde were quantitatively measured for the reaction of pinonaldehyde with OH. The formaldehyde yield was determined to be $1.52 \pm 0.56$, significantly higher than the yield measured in this work. The acetone yield in Noziére et al. (1999) lies with $0.15 \pm 0.07$ in the range of the acetone yield determined here for the times of the experiment when pinonaldehyde is the dominant OH reactant. In the oxidation scheme of pinonaldehyde acetone and formaldehyde are typically formed together so that similar yields would be expected. The high HCHO yield measured by Noziére et al. (1999) can be partially explained by additional fast photolysis of pinonaldehyde and possibly other products by the $254\,nm$ lamps used to generate OH by photodissociaton of $H_2O_2$ (Fantechi et al., 2002).

## 5   Summary and conclusions

The photooxidation of pinonaldehyde was investigated under natural sunlight at low NO concentrations (< 0.2 ppbv) in the presence and absence of an OH scavenger. Two experiments were conducted with maximum pinonaldehyde concentrations of $16.5\,ppbv$ (with OH scavenger) and $6.5\,ppbv$ (without OH scavenger). Measured times series were compared to model calculations based on the recent version of the Master Chemical Mechanism (version 3.3.1).

Model results show that the pinonaldehyde consumption is underestimated in the experiment with OH scavenger. In contrast, the concentration of the measured products acetone and formaldehyde is overestimated by 60 % and 70 %, respectively. The observed decay of pinonaldehyde requires a quantum yield of 0.9 for the photolysis reaction. Previous investigations of the

quantum yield determined lower yields of 0.15 (Moortgat et al., 2002) and $\leq 0.4$ (Jaoui and Kamens, 2003). However, the solar actinic flux could not accurately be determined in these other chamber studies and large corrections for wall loss were applied. Calculations using the measured absorption spectrum (Hallquist et al., 1997) and a quantum yield of 0.9 give photolysis frequencies, which are a factor of 3.5 times higher than values calculated by the parameterization implemented in the MCM, so that photolysis of pinonaldehyde is significantly underestimated, if this parameterization is applied.

Similarly, the pinonaldehyde consumption is underestimated by the MCM model in the experiment, where the pinonaldehyde consumption is dominated by its reaction with OH radicals. Implementing the measured photolysis frequency improves model-measurement agreement. The remaining discrepancy is caused by underestimated OH radical concentrations leading to a slower pinonaldehyde consumption. Constraining $HO_2$ model concentrations to the measurements brings OH concentrations in model and measurement into agreement. As a consequence, also the pinonaldehyde concentration profile is reproduced within the measurement uncertainty. The closed OH budget indicates that a $HO_2$ source is missing in this mechanism. The additional $HO_2$ source would be at least half the rate at which pinonaldehyde is consumed. $HO_2$ would therefore need to be reproduced much faster than current chemical models suggest in one of the major oxidation pathways. Because a large fraction of $RO_2$ radicals (45-80 %) react with $HO_2$, potential reaction pathways that do not lead to the formation of hydroxyperoxide but reform radicals have the potential to contribute the regeneration of $HO_2$. If fast uni-molecular $RO_2$ reactions existed that could compete with $RO_2 + NO$ and $RO_2 + HO_2$ reactions, they could also add to additional $HO_2$ production. Nevertheless, a fast degradation of first-generation products species forming $HO_2$ shows a better agreement with measured $HO_2$ concentration time profiles rather than reactions of $RO_2$ species.

The yield of formaldehyde in the pinonaldehyde photolysis with OH scavenger present is determined to be $0.18 \pm 0.20$. No acetone formation is observed. Model calculations based on the MCM constrained with the measured photolysis frequency overestimate formaldehyde and acetone concentrations by a factor of approximately 3. In the experiment with OH the yields of acetone and formaldehyde increase over the course of the experiment from $(0.2 \pm 0.1)$ to $(0.3 \pm 0.1)$ and from $(0.15 \pm 0.2)$ to $(0.45 \pm 0.2)$ respectively. The increasing yields indicate that both species are also formed by the subsequent chemistry of products formed in the first reaction steps.

Modifications of the degradation mechanism proposed by Fantechi et al. (2002), including a new product distribution and additional products for the initial attack of OH, reproduce measured acetone and formaldehyde concentrations within their uncertainty as long as the reaction with OH is supressed. In the experiment with OH, the model-measurement agreement for both species decrease after implementing the modifications by Fantechi et al. (2002). This indicates that the pathways relevant when OH is dominating the fate of pinonaldehyde lack sources of acetone and formaldehyde in this case.

Field campaigns in environments dominated by monoterpene emissions like the Bio-hydro-atmosphere interactions of Energy, Aerosols, Carbon, H2O, Organics, and Nitrogen–Rocky Mountain Organic Carbon Study (BEACHON-ROCS; Kim et al., 2013) or the Hyytiälä United Measurements of Photochemistry and Particles in Air — Comprehensive Organic Precursor Emission and Concentration study (HUMPPA-COPEC; Hens et al., 2014) showed that OH and $HO_2$ radical concentrations were underestimated in model calculations by up to a factor of 2.5. Constraining modeled $HO_2$ concentrations to measurements allowed reproducing OH radical concentrations. In addition, also chamber studies on the photooxidation of $\alpha$-pinene (Rolletter

et al., 2019) and $\beta$-pinene (Kaminski et al., 2017) confirmed that the current $\alpha$ and $\beta$-pinene mechanisms lack $HO_2$ sources. It is currently unknown, if the missing source is part of reactions forming first generation products or of the subsequent chemistry of the degradation products. Here, it is shown that also in the mechanism of the photoxodidation of pinonaldehyde, a degradation product of $\alpha$-pinene, a $HO_2$ source is missing. However, the findings here cannot explain the discrepancies observed in the $\alpha$-pinene chamber experiments and field campaigns, because the pinonaldehyde yield in the $\alpha$-pinene degradation is rather small (5 %, Rolletter et al. (2019)). Nevertheless, this result is an example for a second-generation species that produces significantly more $HO_2$ than suggested in current chemical models. Further experiments will be required to investigate, if also other oxidation products from the degradation of monoterpenes could explain observations of missing $HO_2$ sources.

*Data availability.*  Data of the experiments in the SAPHIR chamber used in this work is available on the EUROCHAMP data homepage: https://doi.org/10.25326/G53K-WH75 (17.07.2014) and https://doi.org/10.25326/887C-F682 (18.07.2014)

*Author contributions.*  MR analysed the data and wrote the paper. HF and MK designed the experiments. HF conducted the $HO_x$ radical measurements. BB conducted the radiation measurements. MK and RW were responsible for the GC measurements. RT was responsible for the PTR-TOF-MS measurements. XL was responsible for the HONO measurements and HPD for the DOAS OH data. FR was responsible for the $NO_x$ and $O_3$ data. All co-authors commented on the manuscript.

*Competing interests.*  The authors declare to have no competing interests.

*Acknowledgements.*  This work was supported by the EU Horizon 2020 program Eurochamp2020 (grant agreement no. 730997). This project has received funding from the European Research Council (ERC) under the European Union's Horizon 2020 research and innovation programme (SARLEP grant agreement No. 681529) The authors thank Luc Vereecken for his help with the 4-hydroxynorpinonaldehyde mechanism.

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

**Table 1.** Instrumentation for radical and trace gas detection during the pinonaldehyde oxidation experiments.

| | Technique | Time Resolution | $1\sigma$ Precision | $1\sigma$ Accuracy |
|---|---|---|---|---|
| OH | DOAS[a] (Dorn et al., 1995a; Hausmann et al., 1997; Schlosser et al., 2007) | 205 s | $0.8 \times 10^6$ cm$^{-3}$ | 6.5 % |
| OH | LIF[b] (Holland et al., 2003; Fuchs et al., 2012) | 47 s | $0.3 \times 10^6$ cm$^{-3}$ | 13 % |
| HO$_2$ | LIF[b] (Fuchs et al., 2011) | 47 s | $1.5 \times 10^7$ cm$^{-3}$ | 16 % |
| NO | Chemiluminescence (Rohrer and Brüning, 1992) | 180 s | 4 pptv | 5 % |
| NO$_2$ | Chemiluminescence (Rohrer and Brüning, 1992) | 180 s | 2 pptv | 5 % |
| O$_3$ | UV absorption (Ansyco) | 10 s | 5 ppbv | 5 % |
| pinonaldehyde, acetone | PTR-TOF-MS[c] (Lindinger et al., 1998) | 30 s | 15 pptv | 6 % |
| acetone | GC-FID[d] (Kaminski et al., 2017) | 30 min | 20 pptv | 5 % |
| HCHO | Hantzsch monitor (AeroLaser) | 120 s | 20 pptv | 5 % |
| HCHO | DOAS[a] | 100 s | 20 % | 10 % |
| HONO | LOPAP[e] (Li et al., 2014) | 300 s | 1.3 pptv | 13 % |
| Photolysis frequencies | Spectroradiometer (Bohn et al., 2005) | 60 s | 10 % | 10 % |

[a] Differential Optical Absorption Spectroscopy

[b] Laser Induced Fluorescence

[c] Proton-Transfer-Reaction Time-Of-Flight Mass-Spectrometry

[d] Gas Chromatography Flame-Ionization-Detector

[e] Long-Path-Absorption-Photometer

**Table 2.** Overview of different model calculations.

| Model run | Model | $j_{pinonaldehyde}$ | $HO_2$ |
|---|---|---|---|
| MCM | $MCM^a$ | $MCM^b$ | calculated |
| MCM_a | $MCM^a$ | exp.$^c$ | calculated |
| MCM_b | $MCM^a$ | exp.$^c$ | constrained |
| FAN_a | Fantechi et al.$^d$ | exp.$^c$ | calculated |
| FAN_b | Fantechi et al.$^d$ | exp.$^c$ | constrained |
| S1 | like FAN_a, with additional $RO_2 \rightarrow HO_2$ $(0.1\,s^{-1})$ | | |
| | for $RO_2$ = C96CO3, FAN_D1, PINALO2, and FAN_G1 | | |
| S1_mod | like FAN_a, with additional $RO_2 \rightarrow HO_2$ $(0.1\,s^{-1})$ | | |
| | for $RO_2$ = FAN_D1, PINALO2, and FAN_G1 | | |
| S1_mod_hv | like FAN_a, with additional $RO_2 \rightarrow RO_2$isom $(0.1\,s^{-1})$ | | |
| | for $RO_2$ = FAN_D1, PINALO2, and FAN_G1 | | |
| | followed by $RO_2$isom$+h\nu \rightarrow HO_2$ ($j_{glyoxal}$) | | |
| | and $RO_2$isom$+NO \rightarrow HO_2+NO_2$ (KRO2NO$^a$) | | |
| S2 | like FAN_a, with additional photolysis $(0.2 \times j_{NO2})$ | | |
| | of first generation pinonaldehyde products | | |
| | (4-hydroxynorpinonaldehyde, NORPINAL, | | |
| | CO13C4CHO, CO23C4CHO, C818CO) | | |
| S3 | like FAN_a, with subsequent degradation of | | |
| | 4-hydroxynorpinonaldehyde | | |
| $2 \times$ KRO2NO | like FAN_a, with an enhanced reaction rate | | |
| | of $2 \times$ KRO2NO$^a$ for all $RO_2$ + NO reactions | | |

[a] Master Chemical Mechanism v3.3.1

[b] Parametrization used by MCM v3.3.1

[c] Calculated from the measured solar actinic spectrum, using the absorption spectrum by Hallquist et al. (1997) and an estimated effective quantum yield of 0.9

[d] Mechanism by Fantechi et al. (2002) replaces pinonaldehyde chemistry in MCM (see Supplement)

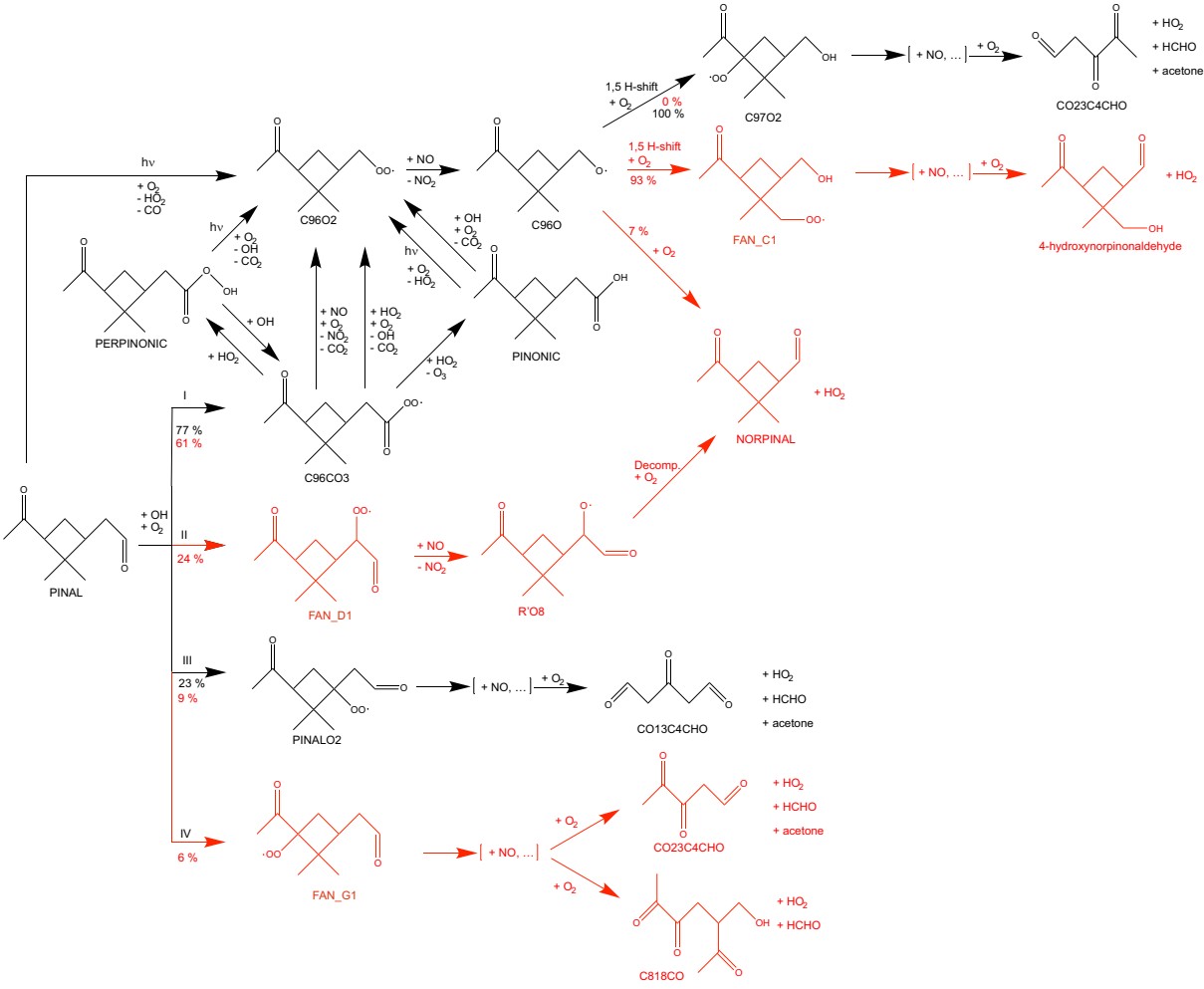

**Figure 1.** Simplified oxidation scheme of pinonaldehyde as described in the MCM (black) and modifications in the model derived by Fantechi et al. (2002) shown in red. [+NO, ...] represents reaction sequences which are initiated by peroxy radical reactions with NO and eventually form carbonyl compounds plus $HO_2$. Possible reactions of $RO_2$ with other $RO_2$ are not shown. $RO_2 + HO_2$ reactions are only shown for the major peroxy radical C96CO3. See text for details.

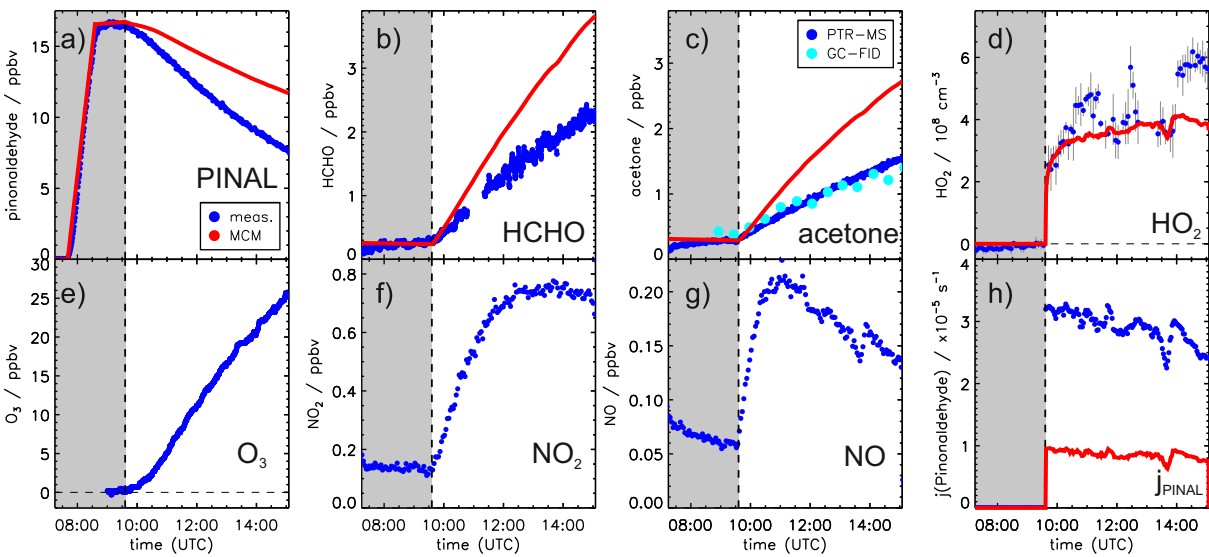

**Figure 2.** Measured and modeled trace gas concentrations and photolysis frequencies during photooxidation of pinonaldehyde in the presence of an OH scavenger. Pinonaldehyde is removed by photolysis, only. Measured $O_3$, $NO_2$ and $NO$ were used as constraints for the model. See text for details of the pinonaldehyde photolysis frequency (h). Grey shaded areas indicate times when the chamber roof was closed.

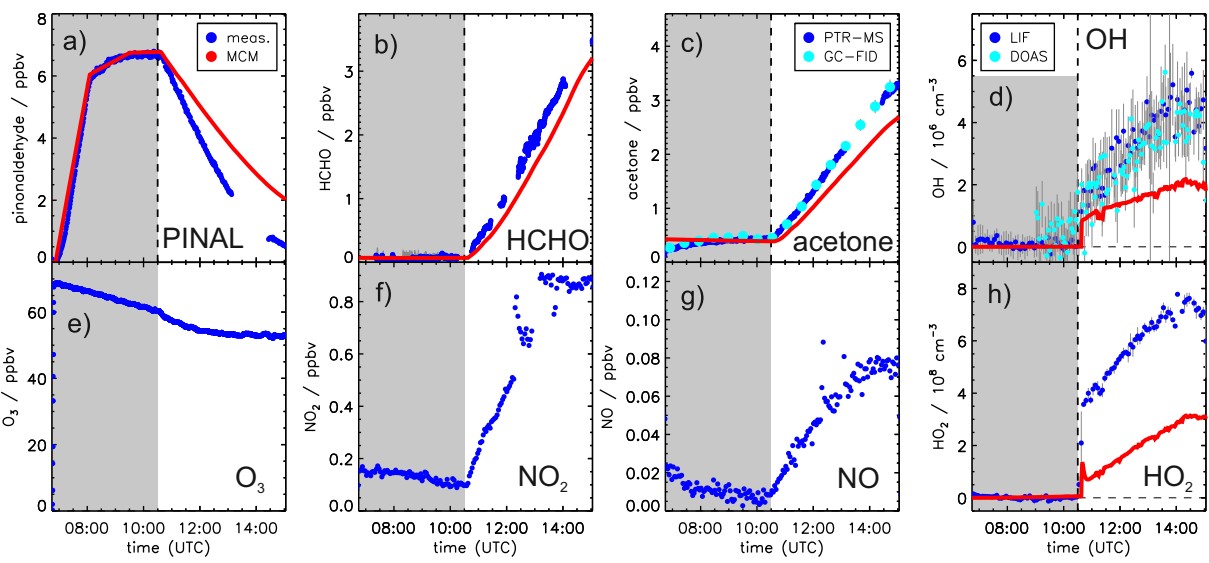

**Figure 3.** Measured and modeled trace gas concentrations during the photooxidation of pinonaldehyde without OH scavenger. In this experiment, pinonaldehyde is removed by photolysis and reaction with OH. Measured $O_3$, $NO_2$ and NO were used as constraints for the model. Grey shaded areas indicate times when the chamber roof was closed.

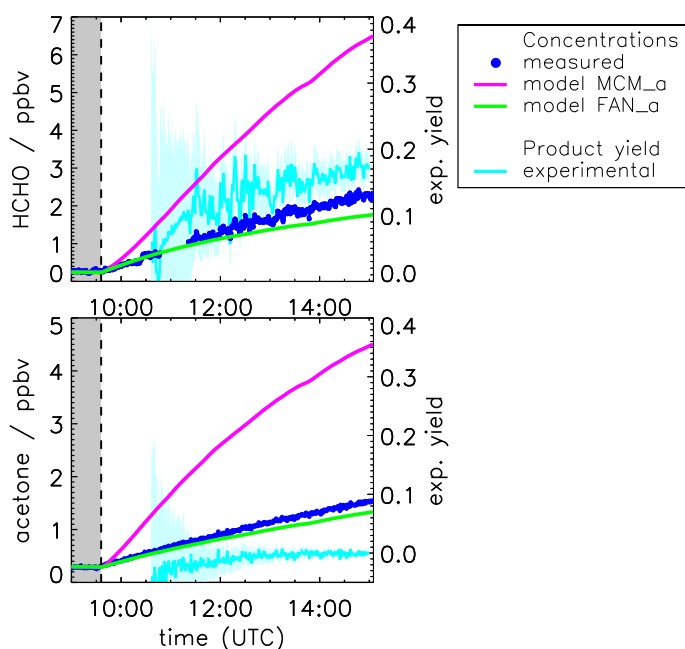

**Figure 4.** Measured and modeled formaldehyde and acetone mixing ratios for the experiment with OH scavenger. All model runs were done with measured photolysis frequencies for pinonaldehyde (see Fig. 6). Model runs include the MCM and the MCM with additions described in Fantechi et al. (2002). In addition, yields calculated from measured time series are shown (see text for details) with the $1\sigma$ error derived from measurements and errors of the applied correction. Colored areas give the uncertainty of this calculation. The additional error caused by the uncertainty of the chamber source is not included here.

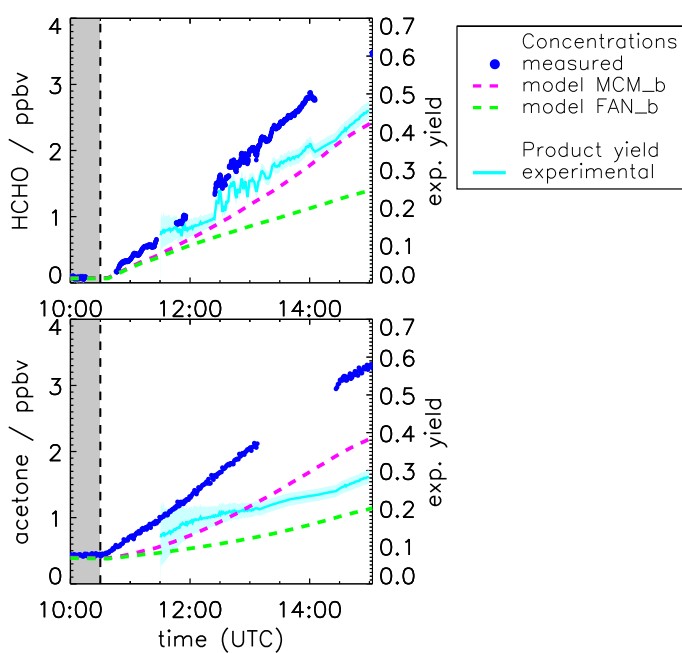

**Figure 5.** Measured and modeled formaldehyde and acetone mixing ratios for the experiment without OH scavenger. All model runs were done with measured photolysis frequencies for pinonaldehyde and with $HO_2$ constrained to measurements (see Fig. 7). Model runs were done using the MCM and the MCM with modifications described in Fantechi et al. (2002). In addition, yields calculated from measured time series are shown (see text for details) with the $1\sigma$ error derived from measurements and errors of the applied correction. Colored areas give the uncertainty of this calculation. The additional error caused by the uncertainty of the chamber source is not included here.

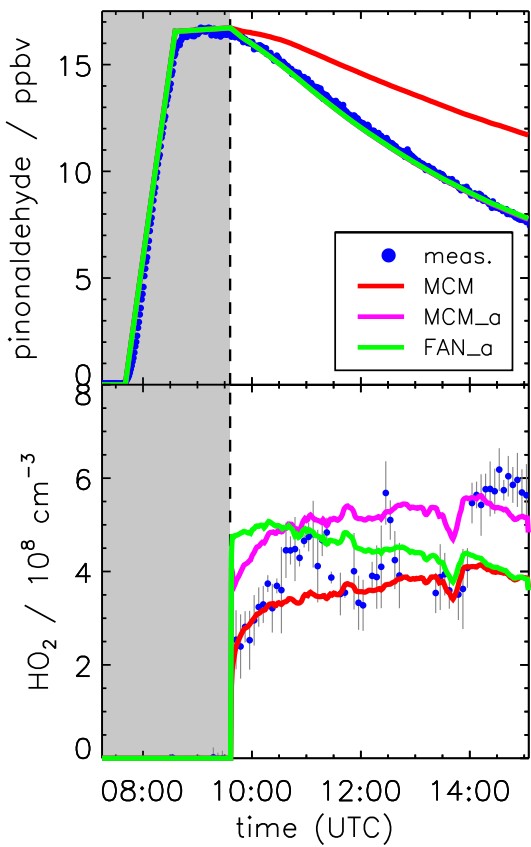

**Figure 6.** Pinonaldehyde and HO$_2$ time series during the experiment with OH scavenger. Model runs were done either using the MCM with parameterization of pinonaldehyde photolysis frequencies (MCM), with measured values for photolysis frequencies (MCM_a, see text for details) or with modifications described in Fantechi et al. (2002) (FAN_a). The pinonaldehyde concentration time profile is the same for both model runs MCM_a and FAN_a. Grey shaded areas indicate times when the chamber roof was closed.

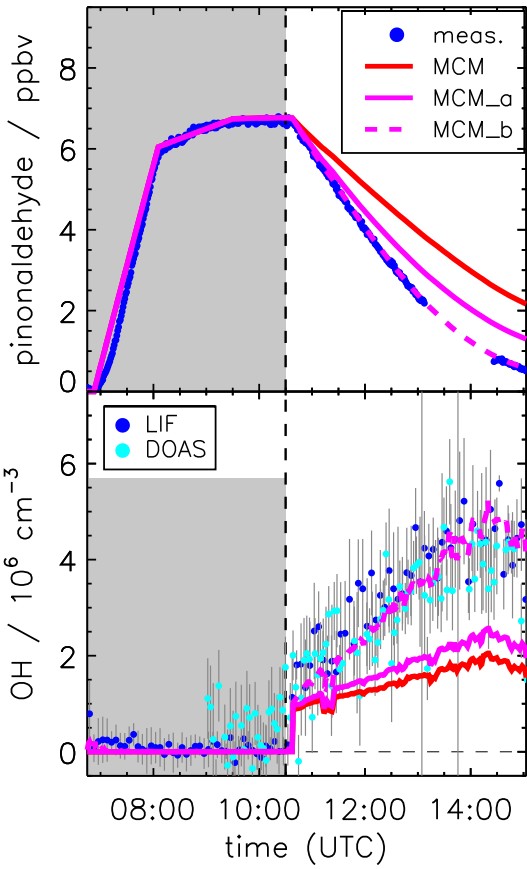

**Figure 7.** Pinonaldehyde and OH time series during the experiment without OH scavenger. Model runs were done either using the MCM with parameterization of pinonaldehyde photolysis frequencies (MCM) or with measured values for photolysis frequencies (MCM_a, see text for details). The model MCM_b was additionally constrained to measured $HO_2$ concentration, resulting in an agreement between modeled and measured (LIF and DOAS) OH concentrations. Grey shaded areas indicate times when the chamber roof was closed.

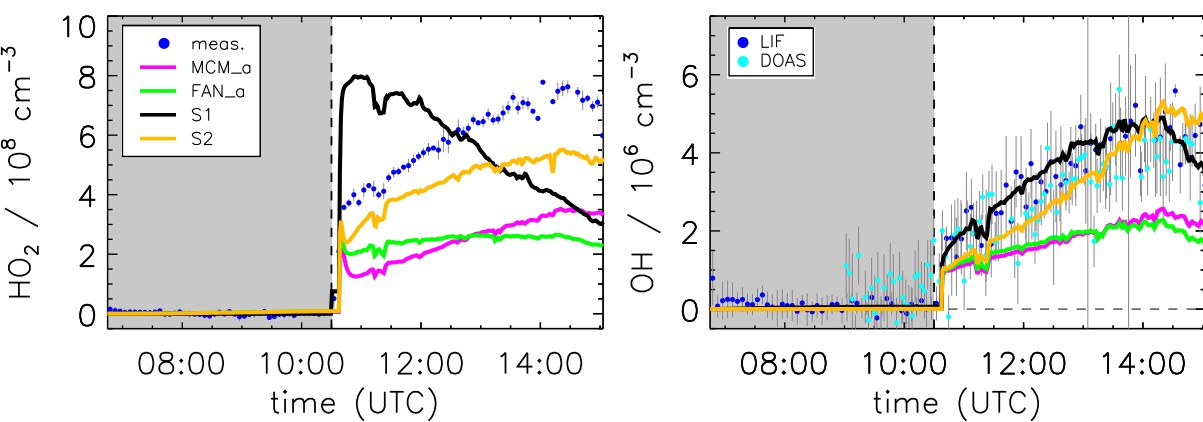

**Figure 8.** Model sensitivity study of the impact of potential additional $HO_2$ formation by unimolecular $RO_2$ reactions (S1) or photolysis of pinonaldehyde oxidation products (S2) compared to the model base case (MCM_a) and the case using the mechanism by Fantechi et al. (2002) (FAN_a). See Table 2 for differences in the model runs. Grey shaded areas indicate times when the chamber roof was closed.