# Peer review of "Photooxidation of pinonaldehyde at ambient conditions investigated in the atmospheric simulation chamber SAPHIR"

_Atmospheric Chemistry and Physics, 2020_

## Referee Comment (RC1) · Anonymous Referee #1 · 2 May 2020

This paper reports results of experiments where pinonaldehyde, a product of atmospheric reactions of biogenic emissions, is photolyzed in a large outdoor environmental chamber in the presence and absence of an OH radical scavenger. The decay of pinonaldehyde, formation of formaldehyde, acetone, and ozone, and levels of OH and HO2 were monitored during the irradiations. However, there were no data on more complex organic products that may give more direct information on the mechanisms. Although results of only one experiment of each type were reported so reproducibility was not tested, this laboratory has a reputation for high quality data in well characterized experiments, and these appear to be no exception. It was found that pinonaldehyde photolyzed significantly more rapidly butyraldehyde, which is what is assumed

in MCM and other mechanisms for higher aldehydes. The product and radical levels observed were not consistent with MCM predictions even after the pinonaldehyde photolysis rate was adjusted, and a more complex mechanism proposed by Fantechi et al (2002) (FAN) gave somewhat better predictions in some respects and somewhat worse in others. No mechanism was presented that was consistent with all the data.

This paper makes a contribution to our understanding of this important biogenic compound that should ultimately improve our ability to model its atmospheric impacts, and should be published. The measurement of the photolysis rate is an important contribution. On the other hand, this work certainly does not resolve uncertainties concerning details of the reaction pathways and the products formed – tests against formaldehyde data and acetone measurements (or lack thereof) are not very definitive because of the many way these small compounds could be formed. The problems fitting the radical data were examined in sensitivity calculations but not really resolved. Nevertheless, these results provide additional data that will be useful for understanding these mechanisms.

Although I recommend that this paper be accepted for publication, I have several comments and suggestions that the authors should consider before finalizing this paper.

Figure 1 shows how the major first-generation photolysis and OH reactions of pinonaldehyde are represented in the MCM and FAN models, and is the only place where the reader is informed of the structure of pinonaldehyde, which many readers won't know without looking it up, and of the various intermediates that are otherwise referenced in the text only with obscure MCM or FAN names. This figure is near the end of the manuscript in the review copy, but it needs to be at the very start of the published version so readers can more easily see the chemistry that is actually being discussed. Either that or have a very small figure near the beginning giving the structures being discussed.

Although the FAN mechanism shown on Figure 1 is an improvement over MCM in

that it considers more of the possible processes, it has some omissions that were not adequately considered in the sensitivity calculations. The paper mentioned that new data indicate that some peroxy radicals undergo unimolecular H-shift reactions, which is overlooked in both MCM and FAN, and the did sensitivity calculations (S1) showing the effect of assuming all those formed in the OH reaction rapidly react forming HO2. However, while the available data and estimates suggest that these H-shifts are fast for peroxy radicals with HCO- groups such as PINALO2, FAN_D1, and Fan_G1, there are no sufficiently labile abstractable H atoms for such reactions of C96CO3, which is formed about 60% of the time in the FAN model, or for the C96O2 radical that it forms. In addition, the reactions subsequent to the initial H-shifts are estimated to form another peroxy radical that converts NO to NO2, and it is not clear that this is included in the sensitivity calculation. If they did it was not stated, and they need to include a supplementary table giving the S1 mechanism, as they do with the FAN mechanism.

It is quite likely that the MCM and FAN mechanisms are underestimating radical input from photolyses of secondary products. If the dicarbonyl parent compound photolyzes faster than expected based on simpler monocarbonyls, that's likely to be true for the many dicarbonyl products as well. They found that increasing photolysis rates to those for glyoxal did not increase radical levels sufficiently, and they had to increase them to that of NO2 (a rather extreme level that is not really reasonable) to get the radicals to the observed levels, and then it greatly overestimates initial HO2 radicals (Model S2). However, if the H-shift isomerizations of peroxy radicals with -HCO groups are indeed as fast as expected, then it is possible that peroxy acids with additional carbonyl groups may well be formed in up to 50% yields. These bifunctional compounds probably photolyze more rapidly than simple peroxy acids, and maybe more rapidly than glyoxal. This possibility would be worth examining.

A major conclusion of this study is that "pinonaldehyde photolyzes faster than MCM predicts". That is true, but it is more to the point, and more meaningful to the photochemical community in general, to convey this as "pinonaldehyde photolyzes faster

than simple aldehydes like butyraldehyde". This is because the MCM website indicates MCM assumes that all compounds like this photolyze as rapidly as buturaldehyde. The fact that this is not true has implications for all mechanisms and photochemistry in general.

I am a bit uncomfortable with using models constrained to fit measured O3, NO, and NO2 when evaluating a mechanism's ability to predict HO2 radical levels. This is because the rapid photostationary state reactions involving O3, NO, and NO2 require that the total rate of all peroxy + NO reactions be approximately equal the differences between the rates of the fast photolysis of NO2 and the fast reaction of O3 with NO, which are much larger in magnitude than the peroxy rates. It seems to me that this might cause small measurement uncertainties in NO, NO2 or O3 to have large effects on artificial radical sources and sinks required to for this constraint to hold. It may be valid the way they did this, but more discussion may be needed. It would be more straightforward and understandable if they could constrain to as few measurements as possible when evaluating mechanisms, and preferably only using reaction conditions such as jNO2. Why not just constrain just one of the reactant predictions, and let the others fall where the mechanism tells you?

The ratio of rate constants used in the model for the reactions of peroxy radicals with HO2 vs. NO may also affect model predictions of HO2. Has this been looked into? These rate constants have not been measured, and estimates based on rate constants for simpler radicals may be an oversimplification.

I have the following minor formatting comments on the figures: (1) Most of the figures are not suitable for presentation in black and white. (2) The species whose concentrations are being plotted should be shown more conspicuously. Right now, this information is only given in very tiny font on the reaction labels. (3) The FAN_a result is not shown in the top plot of Figure 6, probably because it is exactly the same as MCM_a. Perhaps it should be stated in the caption that they are the same. (4) On figure 8, the color shown for MCM on the legend (red) does not match the color on the plots (pink).

---

## Referee Comment (RC2) · Anonymous Referee #2 · 12 May 2020

The paper reports experiments on pinonaldehyde oxidation, conducted in the SAPHIR chamber in Juelich. In one experiment, the oxidation was initiated exclusively by photolysis, in the other, pinonaldehyde reacted primarily with OH. The chamber is highly instrumented and has been very well characterized. The experiments were conducted at low NO and concentrations of pinonaldehyde, HCHO, acetone, NO, NO2, O3, HONO, OH and HO2 were measured, together with the actinic flux outside the chamber, over a period of $\sim$ 4 h. The results were compared with simulations based on the master chemical mechanism (MCM) and on a theoretically based alternative. The results show that there are significant deficiencies in both these mechanisms and suggestions for revisions are made.

[Figure]

The results and the analysis make a significant contribution to improving our understanding of an aspect of the atmospheric oxidation of $\alpha$-pinene – the key pinonaldehyde precursor – and should be published. The authors should consider the following points in a revision of the manuscript.

It is disappointing that only two experiments were conducted, one on photolysis and the other on OH oxidation, so there is no testing of the observations and interpretation via variation, for example, of the initial [NO]. I appreciate that the experiments are complex and resource-intensive, but some comment might be made on the implications of this limitation. In addition, the authors should comment on the strong variations in [HO2] in the photolysis experiment(Figs 2 and 6) which is greater than the experimental uncertainty and is not reproduced in the models. The afternoon decrease is at least partly explained by a decrease in j(pinonaldehyde), but there are other variations also. The variability is much less pronounced in the OH initiation experiments.

It would be helpful if there were an indication of just how important pinonaldehyde is in the oxidation of $\alpha$-pinene. There are a number of routes to pinonaldehyde and their importance depends on conditions, so this is not an easy request to satisfy, but some indication under relevant conditions would be useful, together with an indication of the dominant routes. It is suggested that the yield is small in the field campaigns discussed on p16. Under what conditions is pinonaldehyde production important? Some information on the transmission of the chamber walls and its impact on the spectral distribution should be made. The maximum in the pinonaldehyde spectrum lies below 300 nm, while the absorption has fallen to half its maximum value at $\sim$310 nm, so the photolysis wavelengths and rate could be impacted significantly by the wall transmission and its wavelength dependence.

The photolysis experiment demonstrates that the MCM significantly underestimates the photolysis rate and this is corrected in the paper, which also demonstrates that the yields of acetone and HCHO are substantially overestimated in the MCM. This is addressed by using the mechanism developed by Fantechi et al to describe the

reactions of the oxy radical formed from pinonaldehyde photolysis. As pointed out in the paper, their simulations did not include the subsequent reactions of the main products of this reaction, 4-Hydroxynorpinonaldehyde, which will react further and will also be photolyzed, almost certainly on the timescales of the experiment. It is suggested that it might be a further source of HO2, but it could also be a source of HCHO and acetone. Further discussion, possibly ruling out this possibility, and some suggestion of the likely products is essential. The estimated experimental yield of acetone shown in Fig 4 is initially negative, which is difficult to understand. Some explanation is needed.

The MCM and the Fantechi mechanism cannot explain the yields of HCHO or acetone in the OH initiated oxidation and the paper shows that they seriously underestimate HO2. Two attempts are made to understand the deficiencies, based on isomerization of RO2 species via an autoxidation mechanism and photolysis of intermediates. The latter appears to provide a better explanation. It is disappointing that a more considered analysis isn't given, especially since the senior author of the Fantechi paper leads the theoretical kinetics group at Juelich. I am not suggesting that major calculations are appropriate at this stage, but more informed comments and suggestions of the way forward might be made (e.g. amplifying the comment at lines 8,9 on p 14).

Smaller comments:

What integrator was used in the simulations?

The English needs some attention (e.g. line 12, p12 should be rise (or increase would be better); line 5, p13 replace "MCM like found in" with "MCM, such as the")

Line 15, p15. Should be version 3.3.1.

---

## Author Response (AR1)

*This paper reports results of experiments where pinonaldehyde, a product of atmospheric reactions of biogenic emissions, is photolyzed in a large outdoor environmental chamber in the presence and absence of an OH radical scavenger. The decay of pinonaldehyde, formation of formaldehyde, acetone, and ozone, and levels of OH and HO2 were monitored during the irradiations. However, there were no data on more complex organic products that may give more direct information on the mechanisms. Although results of only one experiment of each type were reported so reproducibility was not tested, this laboratory has a reputation for high quality data in well characterized experiments, and these appear to be no exception. It was found that pinonaldehyde photolyzed significantly more rapidly butyraldehyde, which is what is assumed in MCM and other mechanisms for higher aldehydes. The product and radical levels observed were not consistent with MCM predictions even after the pinonaldehyde photolysis rate was adjusted, and a more complex mechanism proposed by Fantechi et al (2002) (FAN) gave somewhat better predictions in some respects and somewhat worse in others. No mechanism was presented that was consistent with all the data.*

*This paper makes a contribution to our understanding of this important biogenic com-pound that should ultimately improve our ability to model its atmospheric impacts, and should be published. The measurement of the photolysis rate is an important contribution. On the other hand, this work certainly does not resolve uncertainties concerning details of the reaction pathways and the products formed – tests against formaldehyde data and acetone measurements (or lack thereof) are not very definitive because of the many way these small compounds could be formed. The problems fitting the radical data were examined in sensitivity calculations but not really resolved. Nevertheless, these results provide additional data that will be useful for understanding these mechanisms.*

*Although I recommend that this paper be accepted for publication, I have several comments and suggestions that the authors should consider before finalizing this paper.*

We would like to thank the reviewer for reviewing our manuscript and providing helpful comments.

**Comment 1:** *Figure 1 shows how the major first-generation photolysis and OH reactions of pinonaldehyde are represented in the MCM and FAN models, and is the only place where the reader is informed of the structure of pinonaldehyde, which many readers won't know without looking it up, and of the various intermediates that are otherwise referenced in the text only with obscure MCM or FAN names. This figure is near the end of the manuscript in the review copy, but it needs to be at the very start of the published version so readers can more easily see the chemistry that is actually being discussed. Either that or have a very small figure near the beginning giving the structures being discussed.*

**Response:** We agree with the referee that Figure 1 should be placed early in the manuscript as part of the introduction. This will be done during the typesetting process for the final version for ACP.

**Comment 2:** *Although the FAN mechanism shown on Figure 1 is an improvement over MCM in that it considers more of the possible processes, it has some omissions that were not adequately considered in the sensitivity calculations. The paper mentioned that new data indicate that some peroxy radicals undergo unimolecular H-shift reactions, which is overlooked in both MCM and FAN, and the did sensitivity calculations (S1) showing the effect of assuming all those formed in the OH reaction rapidly react forming HO2. However, while the available data and estimates suggest that these H-shifts are fast for peroxy radicals with HCO- groups such as PINALO2, FAN_D1, and Fan_G1, there are no sufficiently labile abstractable H atoms for such reactions of C96CO3, which is formed about 60% of the time in the FAN model, or for the C96O2 radical that it forms. In addition, the reactions subsequent to the initial H-shifts are estimated to form another peroxy radical that converts NO to NO2, and it is not clear that this is included in the sensitivity calculation. If they did it was not stated, and they need to include a supplementary table giving the S1 mechanism, as they do with the FAN mechanism.*

**Response:** In the S1 mechanism only hypothetical isomerization reactions of the initial $RO_2$ radicals (C96CO3, FAN_D1, PINALO2, and FAN_G1) were tested. No subsequent chemistry of the isomerization products was considered. We added an additional table (see Table S1) in the Supplement with reactions considered in the S1 mechanism to clarify this.

In a new sensitivity run (S1_mod) only isomerization of $RO_2$ radicals with a –HCO group (FAN_D1, PINALO2, and FAN_G1) was tested. These $RO_2$ radicals are formed with a yield of 39 %. Therefore, $HO_2$ concentrations in the beginning of the experiment are reduced by a factor of 2 compared to S1 where the isomerization of all initial $RO_2$ leads to the formation of $HO_2$. The reduced $HO_2$ concentrations agree with observations at the start of the photooxidation, but show the same temporal trend as S1.

We add on p14 l19:"However, only FAN_D1, PINALO2, and FAN_G1 have an aldehyde group with a hydrogen that can be easily abstracted (see Supplementary material)." We add the following part as a new section ("Sensitivity study S1 and additional sensitivity tests") to the Supplement:
"In S1 the impact of hypothetical isomerization reactions of all 4 initially formed $RO_2$ radicals on model results was tested. Reactions shown in Tab. S1 were added to the model based on Fantechi et al. (2002). Possible isomerziation reactions in later stages of the mechanism were not tested.

**Table S1.** Overview of added reactions for sensitivity run S1.

| reaction | reaction rate constant |
|---|---|
| C96CO3 $\rightarrow$ HO$_2$ | $0.1\,\mathrm{s}^{-1}$ |
| PINALO2 $\rightarrow$ HO$_2$ | $0.1\,\mathrm{s}^{-1}$ |
| FAN_D1 $\rightarrow$ HO$_2$ | $0.1\,\mathrm{s}^{-1}$ |
| FAN_G1 $\rightarrow$ HO$_2$ | $0.1\,\mathrm{s}^{-1}$ |

However, only PINALO2, FAN_D1, and FAN_G1 have an aldehyde group with a hydrogen that can be easily abstracted. An additional sensitivity study (S1_mod) was performed that includes only isomerization reactions of these 3 $RO_2$ applying the same reaction rates used for S1. Figure S1 shows the calculated $HO_2$ and OH time series together with results from S1, model base case (MCM_a), and modified mechanism by Fantechi et al. (2002) (FAN_a).

[Figure]

**Figure S1.** Model sensitivity studies of the impact of potential additional $HO_2$ formation by unimolecular $RO_2$ reactions of all 4 initial $RO_2$ (S1) and for $RO_2$ radicals with a –HCO group (S1_mod). The sensitivity test S1_mod_hv extends S1_mod by an additional photolysis of isomerization products. In addition, the model base case (MCM_a) and the case using the mechanism by Fantechi et al. (2002) (FAN_a) are shown. Grey shaded areas indicate times when the chamber roof was closed.

PINALO2, FAN_D1, and FAN_G1 are formed with a total yield of 39 %. Therefore, $HO_2$ concentrations in the beginning of the experiment are reduced by a factor of 2 compared to S1 where the isomerization of all initial $RO_2$ leads to the formation of $HO_2$. The reduced $HO_2$ concentrations agree with observations at the start of the photooxidation, but show the same temporal trend as in model run S1 over the course of the experiment. This leads to an increasing model-measurement discrepancy of $HO_2$ concentrations of a factor of up to 3. Consistently, OH concentrations are reduced by a factor of 2 compared to S1."

*Comment 3: It is quite likely that the MCM and FAN mechanisms are underestimating radical input from photolyses of secondary products. If the dicarbonyl parent compound photolyzes faster than expected based on simpler monocarbonyls, that's*

*likely to be true for the many dicarbonyl products as well. They found that increasing photolysis rates to those for glyoxal did not increase radical levels sufficiently, and they had to increase them to that of NO2 (a rather extreme level that is not really reasonable) to get the radicals to the observed levels, and then it greatly overestimates initial HO2 radicals (Model S2). However, if the H-shift isomerizations of peroxy radicals with -HCO groups are indeed as fast as expected, then it is possible that peroxy acids with additional carbonyl groups may well be formed in up to 50% yields. These bifunctional compounds probably photolyze more rapidly than simple peroxy acids, and maybe more rapidly than glyoxal. This possibility would be worth examining.*

**Response:** Following the suggestion of the referee, we did additional sensitivity runs with isomerization of $RO_2$ with a –HCO group followed by photolysis of the isomerization products with a photolysis frequency that is 2 times faster than the photolysis frequency of glyoxal. The photolysis of isomerization products leads to additional $HO_2$ production compared to the sensitivity run with only isomerization (Fig. S1, S1_mod) and reduces the $HO_2$ model-measurement discrepancy especially at later times in the experiment. The sensitivity test is able to reproduce measured OH concentrations within the measurement uncertainty. The result of this sensitivity run (S1_mod_hv) is now included in the Supplement. We add to the manuscript on p15 l3: "Another possibility is that the fast H-shift isomerization of $RO_2$ radicals (see Supplement) leads to the formation of peroxy acids with additional carbonyl functions in high yields. As discussed above, these bi-functional compounds could photolyse faster than currently implemented in the mechanism. A sensitivity test (S1_mod_hv, see Supplement) was performed that includes isomerization of $RO_2$ with a –HCO group. Products are assumed to photolyse with a photolysis rate that is 2 times higher than that of glyoxal. Implementation of these reactions leads to $HO_2$ concentrations that are increased by up to 60 % compared to the sensitivity run that includes only isomerization reactions. Calculated $HO_2$ concentrations underestimate measurements by factor of 2. The sensitivity test reproduces measured OH concentrations within the measurement uncertainty."

*Comment 4: A major conclusion of this study is that "pinonaldehyde photolyzes faster than MCM predicts". That is true, but it is more to the point, and more meaningful to the photochemical community in general, to convey this as "pinonaldehyde photolyzes faster than simple aldehydes like butyraldehyde". This is because the MCM website indicates MCM assumes that all compounds like this photolyze as rapidly as buturaldehyde. The fact that this is not true has implications for all mechanisms and photochemistry in general.*

**Response:** We add on p11 l23: "The pinonaldehyde photolysis is faster than n-butanal because of its two carbonyl functions. This might be valid for other bi-carbonyl compounds that have non-conjugated carbonyl functions, so that the use of the n-butanal photolysis frequency could systematically underestimate the photolysis frequencies of these compounds. However, the high quantum yield close to unity could also be a specific property of pinonaldehyde that might not apply for the photolysis of other bi-carbonyl species."

*Comment 5: I am a bit uncomfortable with using models constrained to fit measured O3, NO, and NO2 when evaluating a mechanism's ability to predict HO2 radical levels. This is be-cause the rapid photostationary state reactions involving O3, NO, and NO2 require that the total rate of all peroxy + NO reactions be approximately equal the differences between the rates of the fast photolysis of NO2 and the fast reaction of O3 with NO, which are much larger in magnitude than the peroxy rates. It seems to me that this might cause small measurement uncertainties in NO, NO2 or O3 to have large effects on artificial radical sources and sinks required to for this constraint to hold. It may be valid the way they did this, but more discussion may be needed. It would be more straightforward and understandable if they could constrain to as few measurements as possible when evaluating mechanisms, and preferably only using reaction conditions such as jNO2. Why not just constrain just one of the reactant predictions, and let the others fall where the mechanism tells you?*

**Response:** One aim of our study is the understanding of $HO_x$ radical concentrations in the pinonaldehyde photooxidation. Because the loss of $HO_2$ in the reaction with NO is a major source of OH, model results could have led to wrong conclusions, if the NO concentration in the model did not match observations. The measurement uncertainties of $O_3$, NO, and $NO_2$ are very small (5 %). Therefore, most accurate results with respect to the turnover rates of radicals are obtained, if measured concentrations are used.

In contrast, not constraining the model to measurements would require an accurate description of all chamber NO sources. The major source of NO in the chamber is photolysis of HONO that is formed on the Teflon wall of the sunlit chamber.

Unfortunately, HONO was only measured in one of these experiments, so that the NO source cannot be accurately described in the model.

We add the reason for the constrains to the manuscript on p8 l12: "These constrains were used because chamber NO sources cannot be modeled accurately and therefore could lead to wrong conclusions in the analysis of turnover rates of radicals."

***Comment 6:*** *The ratio of rate constants used in the model for the reactions of peroxy radicals with HO2 vs. NO may also affect model predictions of HO2. Has this been looked into? These rate constants have not been measured, and estimates based on rate constants for simpler radicals may be an oversimplification.*

**Response:** We performed an additional sensitivity run based on the Fantechi et al. (2002) mechanism with a reaction rate constant for $RO_2 + NO$ (KRO2NO) that is faster by a factor 2 compared to the MCM. Results of this sensitivity runs are shown in Fig. S2. The enhanced reaction rate leads to an increased fraction of $RO_2$ reacting with NO instead of $HO_2$. As a result, $HO_2$ concentrations are enhanced by approximately 50 % compared to the model run FAN_a. However, $HO_x$ concentrations are again underestimated compared to measurements and the sensitivity run cannot reproduce the $HO_2$ concentration time behaviour from observations.

[Figure]

**Figure S2.** Model sensitivity run with modified reaction rate of 2xKRO2NO. The modified reaction rate constant was applied to all $RO_2$ that were introduced by the model modifications based on Fantechi et al. (2002). Grey shaded areas indicate times when the chamber roof was closed.

Figure S2 is added to the Supplement. We add to the manuscript on p13 l21: "Modeled $HO_2$ concentrations could be affected by the use of general reaction rate constants for reactions of $RO_2$ with $HO_2$ and NO (KRO2HO2 and KRO2NO), respectively. This might be an oversimplification for highly functionalized compounds. A sensitivity test (see Supplement) with an enhanced reaction rate for $RO_2 + NO$ reactions of $2\times$ KRO2NO in the modified mechanism by Fantechi et al. (2002) was performed. As a result, the fraction of $RO_2$ reacting with NO instead of $HO_2$ is increased. This leads to an enhanced $HO_2$ concentration of approximately 50 % compared to the model run FAN_a. However, $HO_x$ concentrations are again underestimated compared to measurements and the sensitivity run cannot reproduce the $HO_2$ concentration time behaviour from observations."

***Minor comments:*** *I have the following minor formatting comments on the figures:*
*(1) Most of the figures are not suitable for presentation in black and white.*
Atmospheric Chemistry and Physics is an online journal providing coloured PDF documents. Therefore, coloured figures are very common. We do not have the feeling that there is need to optimize the figures for black and white printing.

*(2) The species whose concentrations are being plotted should be shown more conspicuously. Right now, this information is only given in very tiny font on the reaction labels.*
The names of species shown in Figure 2 and 3 were added in larger font on each sub-plot. The size of the axis labelling was

increased in Figures 4-7.

*(3) The FAN_a result is not shown in the top plot of Figure 6, probably because it is exactly the same as MCM_a. Perhaps it should be stated in the caption that they are the same.*
We changed the figure as suggested.

*(4) On figure 8, the color shown for MCM on the legend (red) does not match the color on the plots (pink).*
We changed the figure to better match colors.

**Anonymous Referee #2**

*The paper reports experiments on pinonaldehyde oxidation, conducted in the SAPHIR chamber in Juelich. In one experiment, the oxidation was initiated exclusively by photolysis, in the other, pinonaldehyde reacted primarily with OH. The chamber is highly instrumented and has been very well characterized. The experiments were conducted at low NO and concentrations of pinonaldehyde, HCHO, acetone, NO, NO2, O3, HONO,OH and HO2 were measured, together with the actinic flux outside the chamber, over a period of ∼4 h. The results were compared with simulations based on the master chemical mechanism (MCM) and on a theoretically based alternative. The results show that there are significant deficiencies in both these mechanisms and suggestions for revisions are made.*
*The results and the analysis make a significant contribution to improving our under-standing of an aspect of the atmospheric oxidation of α-pinene – the key pinonaldehyde precursor – and should be published. The authors should consider the following points in a revision of the manuscript.*

We would like to thank the reviewer for reviewing our manuscript and providing helpful comments.

***Comment 1:*** *It is disappointing that only two experiments were conducted, one on photolysis and the other on OH oxidation, so there is no testing of the observations and interpretation via variation, for example, of the initial [NO]. I appreciate that the experiments are complex and resource-intensive, but some comment might be made on the implications of this limitation. In addition, the authors should comment on the strong variations in [HO2] in the photolysis experiment(Figs 2 and 6) which is greater than the experimental uncertainty and is not reproduced in the models. The afternoon decrease is at least partly explained by a decrease in j(pinonaldehyde), but there are other variations also. The variability is much less pronounced in the OH initiation experiments.*
**Response:** Pinonaldehyde has a low vapour pressure and is a very sticky compound. Therefore, it was complicated to transfer pinonaldehyde into the chamber. In total 4 experiments were performed. Unfortunately, only in 2 experiments shown in this manuscript the transfer of pinonaldehyde into the chamber was successful. In general, experiments in the SAPHIR chamber are very time-consuming and require a large team to measure trace gas concentrations and analyse data. As much as we agree that a larger set of experiments were beneficiary, we could not perform more experiments at different conditions for this work. We nevertheless think that results are important to report.
We add in the manuscript, p12 l18 to discuss the variations of $HO_2$ in the photolysis experiment: "Unfortunately, between 11:30 and 14:00 experimental problems occurred in the $HO_2$ measurements. Neither NO measurements nor photolysis frequencies showed any features that could explain the decrease in the $HO_2$ concentration. The exact reason of the $HO_2$ variations remains unclear and the uncertainty of $HO_2$ measurements is likely higher (50 %) for this period."

***Comment 2:*** *It would be helpful if there were an indication of just how important pinonaldehyde is in the oxidation of α-pinene. There are a number of routes to pinonaldehyde and their importance depends on conditions, so this is not an easy request to satisfy, but some indication under relevant conditions would be useful, together with an indication of the dominant routes. It is suggested that the yield is small in the field campaigns discussed on p16. Under what conditions is pinonaldehyde production important? Some information on the transmission of the chamber walls and its impact on the spectral distribution should be made. The maximum in the pinonaldehyde spectrum lies below 300 nm, while the absorption has fallen to half its maximum*

*value at ~310 nm, so the photolysis wavelengths and rate could be impacted significantly by the wall transmission and its wavelength dependence.*

**Response:** Previous laboratory studies reported a broad range of pinonaldehyde yields (5-87 %, see Rolletter et al., 2019, and references therein) in the photooxidation of $\alpha$-pinene. In the $\alpha$-pinene photooxidation by OH, initially 3 different $RO_2$ are formed and 2 of them eventually form pinonaldehyde. As currently implemented in the MCM pinonaldehyde is formed with a total yield of 84 %. In contrast, a theory based study by Vereecken et al. (2007) suggests a different branching ratio of initial $RO_2$ and additional reaction channels which lead in total to pinonaldehyde yields of 60 % for low atmospheric NO conditions ($\leq 1$ ppbv NO). For laboratory conditions with high NO mixing ratios ($\geq 50$ ppbv NO), the pinonaldehyde yield of their mechanism is reduced to 36 %. Our previous study (Rolletter et al., 2019) showed that further adjustment of the initial $RO_2$ branching ratio in a mechanism based on Vereecken et al. (2007) was necessary to explain the low pinonaldehyde yield of 5 % for conditions similar to the experiments discussed here ($\leq 0.3$ ppbv NO). A similar change in $RO_2$ branching ratios was found in an experimental study by Xu et al. (2019). The current lack of ambient measurement data makes it difficult to explain when the pinonaldehyde formation becomes important.

We add on p3 l19: "In the $\alpha$-pinene photooxidation, initially 3 different peroxy radicals ($RO_2$) are formed and 2 of them eventually form pinonaldehyde."

In addition, we add on p3 l30: "As currently implemented in the Master Chemical Mechanism (MCM, 2017; Jenkin et al., 1997; Saunders et al., 2003) pinonaldehyde is formed with a total yield of 84 %. In contrast, a theory based study by Vereecken et al. (2007) suggested a different branching ratio of initial $RO_2$ and additional reaction channels which lead in total to pinonaldehyde yields of 60 % for low atmospheric NO conditions ($\leq 1$ ppbv NO). Our previous study (Rolletter et al., 2019) showed that further adjustment of the initial $RO_2$ branching ratio in a mechanism based on Vereecken et al. (2007) was necessary to explain the low measured pinonaldehyde yield of 5 % for conditions similar to the experiments discussed here. A similar change in $RO_2$ branching ratios was found in an experimental study by Xu et al. (2019)."

Compared to outdoor conditions actinic flux densities and photolysis frequencies inside the chamber are lower by about 40 % mostly because of shading effects and the transmission of the chamber film. The wavelength dependence of the transmittance of the Teflon was investigated in the laboratory and is part of the model used to calculate photolysis frequencies inside the chamber (Bohn and Zilken, 2005). However, the change of the spectrum is minor, i.e. the spectral distribution inside the chamber is very similar to outside.

We rephrase on p7, l6: "The direct and diffuse actinic flux densities are used as input for a model which calculates mean chamber spectra by taking into account the time-dependent effects of shadings of the chamber steel frame and the transmittance of the Teflon film which is > 0.8 in the complete solar spectral range (Bohn and Zilken, 2005)."

***Comment 3:*** *The photolysis experiment demonstrates that the MCM significantly underestimates the photolysis rate and this is corrected in the paper, which also demonstrates that the yields of acetone and HCHO are substantially overestimated in the MCM. This is addressed by using the mechanism developed by Fantechi et al to describe the reactions of the oxy radical formed from pinonaldehyde photolysis. As pointed out in the paper, their simulations did not include the subsequent reactions of the main products of this reaction, 4-Hydroxynorpinonaldehyde, which will react further and will also be photolyzed, almost certainly on the timescales of the experiment. It is suggested that it might be a further source of HO2, but it could also be a source of HCHO and acetone. Further discussion, possibly ruling out this possibility, and some suggestion of the likely products is essential. The estimated experimental yield of acetone shown in Fig 4 is initially negative, which is difficult to understand. Some explanation is needed.*

**Response:** Corrections of acetone concentrations were applied to experimentally derive the product yield. This includes losses by the reaction with OH and by dilution. Negative yields can arise when the chamber source parameterisation overestimates the acetone formation. This adds to the uncertainty of the yield calculation. The main uncertainty of the yield (20 %) is indeed caused by the uncertainty of the chamber source in addition to the uncertainty that is calculated from the precision of data shown in Fig. 4.

We changed the caption of Figures 4 & 5 to mention that the additional uncertainty from the chamber source is not shown. We add on p11 l34: "Initial acetone yield values are negative because of the high uncertainty in the corrections that are applied in the yield calculation. In the beginning of an experiment, only small amounts of products are formed which leads to a large uncertainty, so that negative values are not significant."

In our response to comment 4 more information is given about potential products in the degradation of 4-hydroxinorpinon-aldehyde. A sensitivity study was performed to test if the channels II, III, and IV of the reaction of pinonaldehyde + OH have the potential to close the gap between modeled and measured acetone and formaldehyde concentrations in the OH photooxidation experiment. Results can be seen in Fig. S3. The additional acetone and HCHO sources can reproduce observations in the first half of the experiment within the measurement uncertainty when contributions from OH reactions of product species are small. In later stages of the experiment, acetone and formaldehyde concentrations are underestimated by the sensitivity model run. Additional acetone and HCHO formation from further degradation of oxidation products not included in the MCM could explain the model-measurement discrepancy. In the photolysis experiment with OH scavenger present, measured acetone and formaldehyde concentrations could be reproduced by the model based on Fantechi et al. (2002). In this case, additional 4-hydroxynorpinonaldehyde photolysis forming acetone and HCHO would lead to an overestimation of modeled acetone and HCHO concentrations.

[Figure]

**Figure S3.** Measured and modeled formaldehyde and acetone mixing ratios for the experiment without OH scavenger. All model runs were done with measured photolysis frequencies for pinonaldehyde and with $HO_2$ constrained to measurements. Model runs were done using modifications described in Fantechi et al. (2002). For the model run shown in black an additional HCHO and acetone formation in pathways II, III, and IV is assumed.

Figure S3 and a description of the sensitivity test are added to the Supplement.

*Comment 4: The MCM and the Fantechi mechanism cannot explain the yields of HCHO or acetone in the OH initiated oxidation and the paper shows that they seriously underestimate HO2. Two attempts are made to understand the deficiencies, based on isomerization of RO2 species via an autoxidation mechanism and photolysis of intermediates. The latter appears to provide a better explanation. It is disappointing that a more considered analysis isn't given, especially since the senior author of the Fantechi paper leads the theoretical kinetics group at Juelich. I am not suggesting that major calculations are appropriate at this stage, but more informed comments and suggestions of the way forward might be made (e.g. amplifying the comment at lines 8,9 on p 14).*
**Response:**
We did further test to explain the unknown $HO_2$ source but could not find any better explanations than discussed in the manuscript. The theory derived pinonaldehyde degradation scheme by Fantechi et al. (2002) has 4 competitive channels with a large uncertainty for the site specificity of the initial OH attack. This could affect the products distrubution. Our analysis relies completely on the product distribution that is prescribed by the model. Quantification of specific products would have helped to better constrain the used mechanism. In the photooxidation experiment a huge variety of different products is formed so that the missing $HO_2$ source cannot be easily explained by the subsequent chemistry of one species. For example, a new degradation mechanism for 4-hydroxynorpinonaldehyde deduced from structure–activity relationship (SAR; Kwok and Atkinson, 1995; Vereecken and Peeters, 2009; Vereecken and Nozière, 2020) was tested. 4-hydroxynorpinonaldehyde is formed as major product in the photooxidation experiment with an overall yield of approximately 25 % and no subsequent chemistry was considered in the MCM model and the modified Fantechi mechanism. However, the impact of the tested subsequent chemistry

of 4-hydroxynorpinonaldehyde on the $HO_2$ formation was small ($\leq 10\%$) compared to the modified mechanism by Fantechi et al. (2002).

The effort for theoretical calculations of the chemistry is very high and cannot be done for all experiments that we perform in the chamber. In addition, no products measurements are available that would help to verify results from theoretical calculations. Therefore, the discussion is limited to provide hints, what will need to be looked into in the future.

We add on p15 l10: "No subsequent chemistry of 4-hydroxynorpinonaldehyde is included in the mechanism so far. In the experiment here 4-hydroxynorpinonaldehyde is formed with an overall yield of approximately 25%. 4-hydroxynorpinonaldehyde is highly functionalized and $RO_2$ radicals formed in its degradation could undergo fast isomerization reactions. For a sensitivity run (S3, see Supplement) a mechanism was deduced with the structure–activity relationship (SAR; Kwok and Atkinson, 1995; Vereecken and Peeters, 2009; Vereecken and Nozière, 2020) method. However, the impact of the tested 4-hydroxynorpinonaldehyde degradation scheme on the $HO_2$ formation was small ($\leq 10\%$) compared to the modified mechanism by Fantechi et al. (2002). Unfortunately, no measurements of stable oxidation products other than acetone and HCHO were available. Without further product measurements the whole analysis discussed here relies on product distribution prescribed by the models. Further experiments that measure oxidation products and yields could help to better constrain branching ratios in degradation mechanisms. In addition, theoretical studies could investigate subsequent degradation schemes of major products in more detail."

We add a section to the Supplement describing the tested degradation mechanism of 4-hydroxynorpinonaldehyde: "In the used mechanism based on Fantechi et al. (2002) 4-hydroxynorpinonaldehyde was formed as main product with an overall yield of approximately 25% but no subsequent chemistry was considered in the MCM model and the modified Fantechi mechanism. To investigate if the subsequent chemistry of this product has the potential to partly explain the missing $HO_2$ source a mechanism was deduced from structure–activity relationship (SAR; Kwok and Atkinson, 1995; Vereecken and Peeters, 2009; Vereecken and Nozière, 2020).

[Figure]

**Figure S4.** Structure of 4-hydroxynorpinonaldehyde and C-atom labeling.

The 4-hydroxynorpinonaldehyde structure is shown in Fig. S4. Reaction rate constants for the H-abstraction by OH were estimated based on Kwok and Atkinson (1995) and are shown in Table S2.

**Table S2.** Reaction rate constants for H-abstraction by OH for different carbon atoms based on Kwok and Atkinson (1995).

| C-atom | reaction rate constant | fraction |
|---|---|---|
| c | $1.69 \times 10^{-11}$ cm$^3$ s$^{-1}$ | 79% |
| d | $3.27 \times 10^{-12}$ cm$^3$ s$^{-1}$ | 15% |
| f | $5.43 \times 10^{-13}$ cm$^3$ s$^{-1}$ | 3% |
| g | $2.62 \times 10^{-13}$ cm$^3$ s$^{-1}$ | 1% |
| h | $5.43 \times 10^{-13}$ cm$^3$ s$^{-1}$ | 3% |

A simplified mechanism of the subsequent degradation of 4-hydroxynorpinonaldehyde is shown in Fig. S5. An overview of added reactions is shown in Table S3. Reaction rates were based on Vereecken and Peeters (2009); Vereecken and Nozière

[Figure]

**Figure S5.** Simplified mechanism of the subsequent degradation of 4-hydroxynorpinonaldehyde. The mechanism is deduced from SAR. For details see text. $RO_2 + HO_2$ reactions and $RO_2 + NO$ reactions that form nitrates are not shown.

(2020). Only the main reaction branches ($\geq 5\%$) were investigated. The mechanism was constructed using SAR described in Jenkin et al. (1997). For all $RO_2 + NO$ reactions the standard reaction rate from MCM (KRO2NO) and an organic nitrate yield of 23 % was used. $RO_2 + HO_2$ reactions were included as described for the modified mechanism based on Fantechi et al. (2002). The photolysis frequency of pinononaldehyde was used for the photolysis of formed hydroperoxides (ROOH).

H-abstraction by OH mainly occurs at the aldehyde group forming the peroxy radical C1. After a rapid $CO_2$ elimination, C1 forms C2 and C3 in equal amounts. C2 can undergo an 1,5 H-shift to form a stable hydroperoxy compound (C5) and $HO_2$. Alternatively, C2 can form the alkoxy radical C4 after reaction with NO. Similarly, C3 forms the alkoxy radical C12. Ring-opening of the 4-membered ring in both C4 and C12 leads to the formation of a peroxy radical C6. Subsequently, the main fraction (approximately 90 %) rearranges after an 1,6 H-shift to C9. C9 either undergoes an 1,6 H-shift forming C10 or forms an alkoxy radical that further decomposes to a stable product and $HO_2$.

A sensitivity run (S3) using the degradation scheme of 4-hydroxynorpinonaldehyde was performed and results can be seen in Fig. S6. The modifications have only a small effect on $HO_2$ and OH concentrations. In the second half of the experiment the degradation of pinonaldehyde oxidation products becomes more relevant and additional $HO_2$ is formed by the 4-hydroxynorpinonaldehyde degradation scheme. However, the effect on the $HO_2$ concentration is small and $HO_2$ concentrations are increased by up to 10 % compared to FAN_a."

[Figure]

**Figure S6.** Model sensitivity study of the impact of a 4-hydroxynorpinonaldehyde degradation mechanism (S3) compared to the model base case (MCM_a) and the case using the mechanism by Fantechi et al. (2002) (FAN_a). Grey shaded areas indicate times when the chamber roof was closed.

**Table S3.** Extended mechanism for the further degradation of 4-hydroxynorpinonaldehyde used for sensitivity test S3. For details see text. All nitrate species are lumped as one species RNO3.

| reaction | reaction rate constant |
|---|---|
| NORPINALOH + OH → C1 | $1.69 \times 10^{-11} \, \mathrm{cm^3 \, s^{-1}}$ [a] |
| NORPINALOH + OH → D1 + HO$_2$ | $3.27 \times 10^{-12} \, \mathrm{cm^3 \, s^{-1}}$ [a] |
| NORPINALOH + OH → F1 + HO$_2$ | $5.43 \times 10^{-13} \, \mathrm{cm^3 \, s^{-1}}$ [a] |
| NORPINALOH + OH → G1 + HO$_2$ | $2.62 \times 10^{-13} \, \mathrm{cm^3 \, s^{-1}}$ [a] |
| NORPINALOH + OH → H1 + HO$_2$ | $5.43 \times 10^{-13} \, \mathrm{cm^3 \, s^{-1}}$ [a] |
| NORPINALOH + h$\nu$ → C2 + HO$_2$ | $j_{\mathrm{PINAL}}$ |
| C1 → C2 + CO$_2$ | KDEC [b] |
| C1 → C3 + CO$_2$ | KDEC [b] |
| C1 → prod. + HO$_2$ | $9.16 \times 10^{-2} \, \mathrm{s^{-1}}$ |
| C1 + NO → C1O + NO$_2$ | 0.77*KRO2NO [c] |
| C1 + NO → RNO3 | 0.23*KRO2NO [c] |
| C1 + HO$_2$ → C1OOH | KRO2HO2 [d] |
| C1OOH + OH → C1 | $1.3 \times 10^{-11} \, \mathrm{cm^3 \, s^{-1}}$ |
| C1OOH + h$\nu$ → C1O + OH | $j_{\mathrm{PINAL}}$ |
| C1O → prod. + HO$_2$ | KDEC [b] |
| C2 + NO → C4 + NO$_2$ | 0.77*KRO2NO [c] |
| C2 + NO → RNO3 | 0.23*KRO2NO [c] |
| C2 → C5 + HO$_2$ | $1.3 \times 10^{-2} \, \mathrm{s^{-1}}$ |
| C2 + HO$_2$ → C2OOH | KRO2HO2 [d] |
| C2OOH + OH → C2 | $1.3 \times 10^{-11} \, \mathrm{cm^3 \, s^{-1}}$ |
| C2OOH + h$\nu$ → C4 + OH | $j_{\mathrm{PINAL}}$ |
| C4 → C6 | KDEC [b] |
| C6 → C9 | $2.8 \times 10^{-1} \, \mathrm{s^{-1}}$ |
| C6 + NO → C7 + NO$_2$ | 0.77*KRO2NO [c] |
| C6 + NO → RNO3 | 0.23*KRO2NO [c] |
| C6 + HO$_2$ → C6OOH | KRO2HO2 [d] |
| C6OOH + OH → C6 | $1.3 \times 10^{-11} \, \mathrm{cm^3 \, s^{-1}}$ |
| C6OOH + h$\nu$ → C7 + OH | $j_{\mathrm{PINAL}}$ |
| C7 → C8 + ACETOL | KDEC [b] |
| C8 → prod. + HO$_2$ | KDEC [b] |
| C9 → C10 + HO$_2$ | $6.6 \times 10^{-4} \, \mathrm{s^{-1}}$ |
| C9 + NO → C11 + NO$_2$ | 0.77*KRO2NO [c] |
| C9 + NO → RNO3 | 0.23*KRO2NO [c] |
| C9 + HO$_2$ → C9OOH | KRO2HO2 [d] |
| C9OOH + OH → C9 | $1.3 \times 10^{-11} \, \mathrm{cm^3 \, s^{-1}}$ |
| C9OOH + h$\nu$ → C11 + OH | $j_{\mathrm{PINAL}}$ |
| C11 → C13 + HO$_2$ | KDEC [b] |
| C3 + NO → C12 + NO$_2$ | 0.77*KRO2NO [c] |
| C3 + NO → RNO3 | 0.23*KRO2NO [c] |
| C3 + HO$_2$ → C3OOH | KRO2HO2 [d] |
| C3OOH + OH → C3 | $1.3 \times 10^{-11} \, \mathrm{cm^3 \, s^{-1}}$ |
| C3OOH + h$\nu$ → C12 + OH | $j_{\mathrm{PINAL}}$ |
| C12 → C6 + HO$_2$ | KDEC [b] |

[a] value from Kwok and Atkinson (1995)
[b] value from MCM: KDEC= $1.0 \times 10^6 \, \mathrm{s^{-1}}$ (MCM, 2017)
[c] value from MCM: KRO2NO= $2.7 \times 10^{-12} \exp(360\mathrm{K/T}) \, \mathrm{cm^3 s^{-1}}$ (MCM, 2017)
[d] value from MCM: KRO2HO2= $2.91 \times 10^{-13} \exp(1300\mathrm{K/T}) \, \mathrm{cm^3 s^{-1}}$ (MCM, 2017)

***Minor comments:***

*What integrator was used in the simulations?*

All model calculations were performed with FACSIMILE as solver.

We add this information in the manuscript (p8, l6): "FACSIMILE was used as solver for differential equations in the model calculations."

*The English needs some attention (e.g. line 12, p12 should be rise (or increase wouldbe better); line 5, p13 replace "MCM like found in" with "MCM, such as the")*

Changed as suggested.

*Line 15, p15. Should be version 3.3.1.*

Changed as suggested.

To account for possible $RO_2 + HO_2$ reactions, a reaction scheme based on the reaction $C97O2 + HO_2$ is added for all newly introduced $RO_2$ species not included in the MCM. $RO_2$ form a corresponding hydroxyperoxide (ROOH) that can either react with OH to regenerate the $RO_2$ or photolyse to form the corresponding alkoxy radical (RO) that would be also formed by the reaction of $RO_2 + NO$. The general scheme is shown here for one $RO_2$ as an example:

$$\mathrm{FAN\_C1 + HO_2} \quad \rightarrow \quad \mathrm{FAN\_C1\_HO2} \quad \mathrm{(KRO2HO2)} \tag{R-S1}$$

$$\mathrm{FAN\_C1\_HO2 + OH} \quad \rightarrow \quad \mathrm{FAN\_C1} \quad (1 \times 10^{-11} \mathrm{cm^3 s^{-1}}) \tag{R-S2}$$

$$\mathrm{FAN\_C1\_HO2 + h\nu} \quad \rightarrow \quad \mathrm{R\_O3} \quad (\mathrm{J_{22}}) \tag{R-S3}$$

$$\mathrm{FAN\_C1\_HO2 + h\nu} \quad \rightarrow \quad \mathrm{R\_O3} \quad (\mathrm{J_{41}}) \tag{R-S4}$$

Reaction rate constants were used as stated in the MCM.

**Table S1.** Additional and modified reactions applied to the MCM based on the proposed mechanism by Fantechi et al. (2002). Names are taken from the MCM where existing. Newly introduced species are named either with the prefix "FAN" or "R". All nitrate species are lumped as one species RNO3.

| reaction | reaction rate constant |
|---|---|
| PINAL + OH → C96CO3 | $0.61 \times 5.2 \times 10^{-12} \exp(600K/T) \, cm^3 s^{-1}$ |
| PINAL + OH → FAN_D1 | $0.24 \times 5.2 \times 10^{-12} \exp(600K/T) \, cm^3 s^{-1}$ |
| PINAL + OH → PINALO2 | $0.09 \times 5.2 \times 10^{-12} \exp(600K/T) \, cm^3 s^{-1}$ |
| PINAL + OH → FAN_G1 | $0.06 \times 5.2 \times 10^{-12} \exp(600K/T) \, cm^3 s^{-1}$ |
| C96O → NORPINAL + HO$_2$ | $5.0 \times 10^4 \, s^{-1}$ |
| C96O → FAN_C1 | $6.5 \times 10^5 \, s^{-1}$ |
| FAN_C1 + NO → R_O3 + NO$_2$ | $0.86 \times$ KRO2NO[a] |
| FAN_C1 + NO → RNO3 | $0.14 \times$ KRO2NO[a] |
| R_O3 → FAN_C2 + HCHO | $1.2 \times 10^7 \, s^{-1}$ |
| R_O3 → FAN_C3 | $3.2 \times 10^8 \, s^{-1}$ |
| FAN_C2 + NO → R_O5 +NO$_2$ | $0.91 \times$ KRO2NO[a] |
| FAN_C2 + NO → RNO3 | $0.09 \times$ KRO2NO[a] |
| FAN_C3 → NORPINALOH + HO$_2$ | $2.0 \times 10^3 \, s^{-1}$ |
| R_O5 → FAN_C5 | $0.5 \times$ KDEC[b] |
| R_O5 → FAN_C6 | $0.5 \times$ KDEC[b] |
| FAN_C5 + NO → FAN_C7 + NO$_2$ | $0.75 \times$ KRO2NO[a] |
| FAN_C5 + NO → RNO3 | $0.25 \times$ KRO2NO[a] |
| FAN_C6 + NO → HCHO + HO$_2$ + NO$_2$ | $0.93 \times$ KRO2NO[a] |
| FAN_C6 + NO → RNO3 | $0.07 \times$ KRO2NO[a] |
| FAN_D1 + NO → R_O8 + NO$_2$ | $0.72 \times$ KRO2NO[a] |
| FAN_D1 + NO → RNO3 | $0.28 \times$ KRO2NO[a] |
| R_O8 → NORPINAL + HO$_2$ | KDEC[b] |
| FAN_G1 + NO → FAN_G2 + NO$_2$ | KRO2NO[a] |
| FAN_G2 + NO → R_O13 + NO$_2$ | $0.89 \times$ KRO2NO[a] |
| FAN_G2 + NO → RNO3 | $0.11 \times$ KRO2NO[a] |
| R_O13→ FAN_G3 | $5.0 \times 10^{11} \, s^{-1}$ |
| FAN_G3 + NO → R_O14 + NO$_2$ | KRO2NO[a] |
| R_O14→ FAN_G4 + CO$_2$ | KDEC[b] |
| FAN_G4 + NO → R_O15 + NO$_2$ | $0.86 \times$ KRO2NO[a] |
| FAN_G4 + NO → RNO3 | $0.14 \times$ KRO2NO[a] |
| R_O15→ FAN_G5 | $0.5 \times 1.0 \times 10^5 \, s^{-1}$ |
| R_O15→ FAN_G7 | $0.5 \times 1.0 \times 10^5 \, s^{-1}$ |
| FAN_G5 + NO → R_O16 + NO$_2$ | $0.97 \times$ KRO2NO[a] |
| FAN_G5 + NO → RNO3 | $0.03 \times$ KRO2NO[a] |
| R_O16→ C818CO + HCHO + HO$_2$ | KDEC[b] |
| FAN_G7 + NO → R_O17 + NO$_2$ | $0.97 \times$ KRO2NO[a] |
| FAN_G7 + NO → RNO3 | $0.03 \times$ KRO2NO[a] |
| R_O17→ CO23C4CHO + CH3COCH3 + HO$_2$ | KDEC[b] |

[a] value from MCM: KRO2NO= $2.7 \times 10^{-12} \exp(360K/T) \, cm^3 s^{-1}$ (MCM, 2017)
[b] value from MCM: KDEC= $1.0 \times 10^6 \, s^{-1}$ (MCM, 2017)

**Sensitivity study S1 and additional sensitivity tests**

In S1 the impact of hypothetical isomerization reactions of all 4 initially formed $RO_2$ radicals on model results was tested. Reactions shown in Tab. S2 were added to the model based on Fantechi et al. (2002). Possible isomerziation reactions in later stages of the mechanism were not tested.

**Table S2.** Overview of added reactions for sensitivity run S1.

| reaction | reaction rate constant |
|---|---|
| $C96CO3 \rightarrow HO_2$ | $0.1\,s^{-1}$ |
| $PINALO2 \rightarrow HO_2$ | $0.1\,s^{-1}$ |
| $FAN\_D1 \rightarrow HO_2$ | $0.1\,s^{-1}$ |
| $FAN\_G1 \rightarrow HO_2$ | $0.1\,s^{-1}$ |

However, only PINALO2, FAN_D1, and FAN_G1 have an aldehyde group with a hydrogen that can be easily abstracted. An additional sensitivity study (S1_mod) was performed that includes only isomerization reactions of these 3 $RO_2$ applying the same reaction rates used for S1. Figure S1 shows the calculated $HO_2$ and OH time series together with results from S1, model base case (MCM_a), and modified mechanism by Fantechi et al. (2002) (FAN_a).

[Figure]

**Figure S1.** Model sensitivity studies of the impact of potential additional $HO_2$ formation by unimolecular $RO_2$ reactions of all 4 initial $RO_2$ (S1) and for $RO_2$ radicals with a –HCO group (S1_mod). The sensitivity test S1_mod_hv extends S1_mod by an additional photolysis of isomerization products. In addition, the model base case (MCM_a) and the case using the mechanism by Fantechi et al. (2002) (FAN_a) are shown. Grey shaded areas indicate times when the chamber roof was closed.

These $RO_2$ radicals are formed with a yield of 39 %. Therefore, $HO_2$ concentrations in the beginning of the experiment are reduced by a factor of 2 compared to S1 where the isomerization of all initial $RO_2$ leads to the formation of $HO_2$. The reduced $HO_2$ concentrations agree with observations at the start of the photooxidation, but show the same temporal trend as in model run S1 over the course of the experiment. This leads to an increasing model-measurement discrepancy of $HO_2$ concentrations of a factor of up to 3. Consistently, OH concentrations are reduced by a factor of 2 compared to S1.

Products of the rapid isomerization reaction could be peroxy acids with additional carbonyl functions. As seen for pinonalde-hyde, photolysis frequencies of bi-carbonyl compounds could be generally underestimated in current models. An additional sensitivity test (S1_mod_hv) with isomerization of initially formed $RO_2$ with –HCO group followed by photolysis of the isomerization products with the photolysis frequency of glyoxal was performed. Because the photolysis frequency of glyoxal is slow compared to the reaction rate of the isomerization reaction, the $HO_2$ concentration time series in this sensitivity model run is similar to the sensitivity run with only isomerization (S1_mod). The formation of $HO_2$ is linked to the $RO_2$ concentration in this case and underestimated by the model.

**Sensitivity test of $RO_2 + NO$ reaction rate constants**

[Figure]

**Figure S2.** Model sensitivity run with modified reaction rate of 2xKRO2NO. The modified reaction rate constant was applied to all $RO_2$ that were introduced by the model modifications based on Fantechi et al. (2002). Grey shaded areas indicate times when the chamber roof was closed.

**Sensitivity study S3**

In the used mechanism based on Fantechi et al. (2002) 4-hydroxynorpinonaldehyde was formed as main product with an overall yield of approximately 25 % but no subsequent chemistry was considered in the MCM model and the modified Fantechi mechanism. To investigate if the subsequent chemistry of this product has the potential to partly explain the missing $HO_2$ source a mechanism was deduced from structure–activity relationship (SAR; Kwok and Atkinson, 1995; Vereecken and Peeters, 2009; Vereecken and Nozière, 2020). No theoretical calculations were performed.

**Figure S3.** Structure of 4-hydroxynorpinonaldehyde and C-atom labeling.

The 4-hydroxynorpinonaldehyde structure is shown in Fig. S3. Reaction rate constants for the H-abstraction by OH were estimated based on Kwok and Atkinson (1995) and are shown in Table S3.

**Table S3.** Reaction rate constants for H-abstraction by OH for different carbon atoms based on Kwok and Atkinson (1995).

| C-atom | reaction rate constant | fraction |
|--------|------------------------|----------|
| c | $1.69 \times 10^{-11}\,\mathrm{cm^3\,s^{-1}}$ | 79 % |
| d | $3.27 \times 10^{-12}\,\mathrm{cm^3\,s^{-1}}$ | 15 % |
| f | $5.43 \times 10^{-13}\,\mathrm{cm^3\,s^{-1}}$ | 3 % |
| g | $2.62 \times 10^{-13}\,\mathrm{cm^3\,s^{-1}}$ | 1 % |
| h | $5.43 \times 10^{-13}\,\mathrm{cm^3\,s^{-1}}$ | 3 % |

A simplified mechanism of the subsequent degradation of 4-hydroxynorpinonaldehyde is shown in S4. An overview of added reactions is shown in Tab. **??**. Reaction rates are base on Vereecken and Peeters (2009); Vereecken and Nozière (2020). Only the main reaction branches ($\geq 5\%$) were investigated. The mechanism was constructed according to Jenkin et al. (1997). For all $RO_2 + NO$ reactions the standard reaction rate from MCM (KRO2NO) and an organic nitrate yield of 23 % was used. $RO_2 + HO_2$ reactions were included as described for the modified mechanism based on Fantechi et al. (2002). The photolysis frequency of pinononaldehyde was used for the photolysis of formed hydroperoxides (ROOH).

H-abstraction by OH mainly occurs at the aldehyde group forming the peroxy radical C1. After a rapid $CO_2$ elimination, C1 forms C2 and C3 in equal amounts. C2 can undergo an 1,5 H-shift to form a stable hydroperoxy compound (C5) and $HO_2$. Alternatively, C2 can form the alkoxy radical C4 after reaction with NO. Similarly, C3 forms the alkoxy radical C12. Ring-opening of the 4-membered ring in both C4 and C12 leads to the formation of a peroxy radical C6. Subsequently, the main fraction (approximately 90 %) rearranges after an 1,6 H-shift to C9. C9 either undergoes an 1,6 H-shift forming C10 or forms an alkoxy radical that further decomposes to a stable product and $HO_2$.

A sensitivity run (S3) using the degradation scheme of 4-hydroxynorpinonaldehyde was performed and results can be seen in Fig. S5. The modifications have only a small effect on $HO_2$ and OH concentrations. In the second half of the experiment the degradation of pinonaldehyde oxidation products becomes more relevant and additional $HO_2$ is formed by the 4-hydroxynorpinonaldehyde degradation scheme. However, the effect on the $HO_2$ concentration is small and $HO_2$ concentrations are increased by up to 10 % compared to FAN_a."

[Figure]

**Figure S4.** Simplified mechanism of the subsequent degradation of 4-hydroxynorpinonaldehyde. The mechanism is deduced from SAR. For details see text. $RO_2 + HO_2$ reactions and $RO_2 + NO$ reactions that form nitrates are not shown.

[Figure]

**Figure S5.** Model sensitivity study of the impact of a 4-hydroxynorpinonaldehyde degradation mechanism (S3) compared to the model base case (MCM_a) and the case using the mechanism by Fantechi et al. (2002) (FAN_a). Grey shaded areas indicate times when the chamber roof was closed.

**Table S4.** Extended mechanism for the further degradation of 4-hydroxynorpinonaldehyde used for sensitivity test S3. For details see text. All nitrate species are lumped as one species RNO3.

| reaction | reaction rate constant |
|---|---|
| NORPINALOH + OH → C1 | $1.69 \times 10^{-11}\,\mathrm{cm^3\,s^{-1}}$ [a] |
| NORPINALOH + OH → D1 + HO$_2$ | $3.27 \times 10^{-12}\,\mathrm{cm^3\,s^{-1}}$ [a] |
| NORPINALOH + OH → F1 + HO$_2$ | $5.43 \times 10^{-13}\,\mathrm{cm^3\,s^{-1}}$ [a] |
| NORPINALOH + OH → G1 + HO$_2$ | $2.62 \times 10^{-13}\,\mathrm{cm^3\,s^{-1}}$ [a] |
| NORPINALOH + OH → H1 + HO$_2$ | $5.43 \times 10^{-13}\,\mathrm{cm^3\,s^{-1}}$ [a] |
| NORPINALOH + h$\nu$ → C2 + HO$_2$ | $j_{\mathrm{PINAL}}$ |
| C1 → C2 + CO$_2$ | KDEC[b] |
| C1 → C3 + CO$_2$ | KDEC[b] |
| C1 → prod. + HO$_2$ | $9.16 \times 10^{-2}\,\mathrm{s^{-1}}$ |
| C1 + NO → C1O + NO$_2$ | 0.77*KRO2NO[c] |
| C1 + NO → RNO3 | 0.23*KRO2NO[c] |
| C1 + HO$_2$ → C1OOH | KRO2HO2[d] |
| C1OOH + OH → C1 | $1.3 \times 10^{-11}\,\mathrm{cm^3\,s^{-1}}$ |
| C1OOH + h$\nu$ → C1O + OH | $j_{\mathrm{PINAL}}$ |
| C1O→ prod. + HO$_2$ | KDEC[b] |
| C2 + NO → C4 + NO$_2$ | 0.77*KRO2NO[c] |
| C2 + NO → RNO3 | 0.23*KRO2NO[c] |
| C2 → C5 + HO$_2$ | $1.3 \times 10^{-2}\,\mathrm{s^{-1}}$ |
| C2 + HO$_2$ → C2OOH | KRO2HO2[d] |
| C2OOH + OH → C2 | $1.3 \times 10^{-11}\,\mathrm{cm^3\,s^{-1}}$ |
| C2OOH + h$\nu$ → C4 + OH | $j_{\mathrm{PINAL}}$ |
| C4 → C6 | KDEC[b] |
| C6 → C9 | $2.8 \times 10^{-1}\,\mathrm{s^{-1}}$ |
| C6 + NO → C7 + NO$_2$ | 0.77*KRO2NO[c] |
| C6 + NO → RNO3 | 0.23*KRO2NO[c] |
| C6 + HO$_2$ → C6OOH | KRO2HO2[d] |
| C6OOH + OH → C6 | $1.3 \times 10^{-11}\,\mathrm{cm^3\,s^{-1}}$ |
| C6OOH + h$\nu$ → C7 + OH | $j_{\mathrm{PINAL}}$ |
| C7 → C8 + ACETOL | KDEC[b] |
| C8 → prod. + HO$_2$ | KDEC[b] |
| C9 → C10 + HO$_2$ | $6.6 \times 10^{-4}\,\mathrm{s^{-1}}$ |
| C9 + NO → C11 + NO$_2$ | 0.77*KRO2NO[c] |
| C9 + NO → RNO3 | 0.23*KRO2NO[c] |
| C9 + HO$_2$ → C9OOH | KRO2HO2[d] |
| C9OOH + OH → C9 | $1.3 \times 10^{-11}\,\mathrm{cm^3\,s^{-1}}$ |
| C9OOH + h$\nu$ → C11 + OH | $j_{\mathrm{PINAL}}$ |
| C11 → C13 + HO$_2$ | KDEC[b] |
| C3 + NO → C12 + NO$_2$ | 0.77*KRO2NO[c] |
| C3 + NO → RNO3 | 0.23*KRO2NO[c] |
| C3 + HO$_2$ → C3OOH | KRO2HO2[d] |
| C3OOH + OH → C3 | $1.3 \times 10^{-11}\,\mathrm{cm^3\,s^{-1}}$ |
| C3OOH + h$\nu$ → C12 + OH | $j_{\mathrm{PINAL}}$ |
| C12 → C6 + HO$_2$ | KDEC[b] |

[a] value from Kwok and Atkinson (1995)
[b] value from MCM: KDEC= $1.0 \times 10^{6}\,\mathrm{s^{-1}}$ (MCM, 2017)
[c] value from MCM: KRO2NO= $2.7 \times 10^{-12}\exp(360\mathrm{K/T})\,\mathrm{cm^3 s^{-1}}$ (MCM, 2017)
[d] value from MCM: KRO2HO2= $2.91 \times 10^{-13}\exp(1300\mathrm{K/T})\,\mathrm{cm^3 s^{-1}}$ (MCM, 2017)

**Sensitivity test of additional acetone and HCHO formation by pathways II, III, and IV**

In a sensitivity study it was tested if the pathways II, III, and IV that do not form 4-hydroxynorpinonaldehyde have the potential to explain the missing acetone and formaldehyde formation in the OH oxidation experiment. In a sensitivity test the first reaction step of the pathways II, III, and IV form one molecule of acetone and HCHO each. Results are shown in Fig. S6. The additional acetone and HCHO sources can reproduce observations in the first half of the experiment within the measurement uncertainty when contributions from OH reactions of product species are small. In later stages of the experiment, acetone and formaldehyde concentrations are underestimated by the sensitivity model run. Additional acetone and HCHO formation from further degradation of oxidation products not included in the MCM could explain the model-measurement discrepancy. See the response to comment 4 for more information of potential products of the degradation of 4-hydroxinorpinonaldehyde.

[Figure]

**Figure S6.** Measured and modeled formaldehyde and acetone mixing ratios for the experiment without OH scavenger. All model runs were done with measured photolysis frequencies for pinonaldehyde and with $HO_2$ constrained to measurements. Model runs were done using modifications described in Fantechi et al. (2002). For the model run shown in black an additional HCHO and acetone formation in pathways II, III, and IV is assumed.

---

## Author Response (AR2)

*This paper reports results of experiments where pinonaldehyde, a product of atmospheric reactions of biogenic emissions, is photolyzed in a large outdoor environmental chamber in the presence and absence of an OH radical scavenger. The decay of pinonaldehyde, formation of formaldehyde, acetone, and ozone, and levels of OH and HO2 were monitored during the irradiations. It was found that pinonaldehyde photolyzed significantly more rapidly than assumed in MCM and other mechanisms for higher aldehydes. The product and radical levels observed were not consistent with MCM predictions even after the pinonaldehyde photolysis rate was adjusted, and a more complex mechanism proposed by Fantechi et al (2002) (FAN) gave somewhat better predictions in some respects and somewhat worse in others. A number of sensitivity calculations were conducted to evaluate this, but no mechanism was presented that was consistent with all the data.*

*This is a revised version of a manuscript that I reviewed previously. The significance and scientific quality was rated as excellent, but I did have some comments and suggestions. The revisions sufficiently address my comments, and those of the other reviewer, that I now recommend that it be accepted for publications. I do have one additional suggestion that the authors may consider. In my review I suggested that several possible alternative mechanism options that may affect their observations, and the response was that several new sensitivity calculations were carried out, whose results are given in the Supplement and briefly summarized in the main text. However, not all sensitivity calculations (e.g., "S1_mod", "S1_mod_hv", "2 x KRO2NO, "S3" and "Pathways II-IV") are included in Table 2, the summary of calculations carried out. I suggest that these additional calculations also be listed and summarized in the table. An additional column giving the figure(s) where the results may be seen would also be helpful.*

We would like to thank the reviewer for reviewing our manuscript and providing helpful comments. Missing sensitivity calculations were added to the table:

**Table S1.** Overview of different model calculations.

| Model run | Model | $j_{\text{pinonaldehyde}}$ | $HO_2$ |
|---|---|---|---|
| MCM | MCM[a] | MCM[b] | calculated |
| MCM_a | MCM[a] | exp.[c] | calculated |
| MCM_b | MCM[a] | exp.[c] | constrained |
| FAN_a | Fantechi et al.[d] | exp.[c] | calculated |
| FAN_b | Fantechi et al.[d] | exp.[c] | constrained |
| S1 | like FAN_a, with additional $RO_2 \rightarrow HO_2$ $(0.1\,\text{s}^{-1})$ for $RO_2 =$ C96CO3, FAN_D1, PINALO2, and FAN_G1 | | |
| S1_mod | like FAN_a, with additional $RO_2 \rightarrow HO_2$ $(0.1\,\text{s}^{-1})$ for $RO_2 =$ FAN_D1, PINALO2, and FAN_G1 | | |
| S1_mod_hv | like FAN_a, with additional $RO_2 \rightarrow RO_2\text{isom}$ $(0.1\,\text{s}^{-1})$ for $RO_2 =$ FAN_D1, PINALO2, and FAN_G1 followed by $RO_2\text{isom}+h\nu \rightarrow HO_2$ $(j_{\text{glyoxal}})$ and $RO_2\text{isom}+NO \rightarrow HO_2+NO_2$ (KRO2NO[a]) | | |
| S2 | like FAN_a, with additional photolysis $(0.2 \times j_{NO2})$ of first generation pinonaldehyde products (4-hydroxynorpinonaldehyde, NORPINAL, CO13C4CHO, CO23C4CHO, C818CO) | | |
| S3 | like FAN_a, with subsequent degradation of 4-hydroxynorpinonaldehyde | | |
| $2 \times$ KRO2NO | like FAN_a, with an enhanced reaction rate of $2 \times$ KRO2NO[a] for all $RO_2$ + NO reactions | | |

[a] Master Chemical Mechanism v3.3.1
[b] Parametrization used by MCM v3.3.1
[c] Calculated from the measured solar actinic spectrum, using the absorption spectrum by Hallquist et al. (1997) and an estimated effective quantum yield of 0.9
[d] Mechanism by Fantechi et al. (2002) replaces pinonaldehyde chemistry in MCM (see Supplement)

**Anonymous Referee #3**

*The authors have provided a detailed set of responses to the reviewers' comments and I recommend that the manuscript is suitable for publication. There are some instances where some minor English language editing would be beneficial, and the authors may wish to provide some details of the calibration procedures for HCHO/CH3CHO given the significance of the measurements of these species in the manuscript.*

We would like to thank the reviewer for reviewing our manuscript and providing helpful comments. We added details on the formaldehyde and acetone calibration to the manuscript:

[revised manuscript text omitted]